# Indefinitely stable iron(IV) cage complexes formed in water by air oxidation

Stefania Tomyn[1], Sergii I. Shylin[1,2], Dmytro Bykov[3], Vadim Ksenofontov[2], Elzbieta Gumienna-Kontecka[4], Volodymyr Bon[5,6] & Igor O. Fritsky[1]

In nature, iron, the fourth most abundant element of the Earth's crust, occurs in its stable forms either as the native metal or in its compounds in the $+2$ or $+3$ (low-valent) oxidation states. High-valent iron ($+4$, $+5$, $+6$) compounds are not formed spontaneously at ambient conditions, and the ones obtained synthetically appear to be unstable in polar organic solvents, especially aqueous solutions, and this is what limits their studies and use. Here we describe unprecedented iron(IV) hexahydrazide clathrochelate complexes that are assembled in alkaline aqueous media from iron(III) salts, oxalodihydrazide and formaldehyde in the course of a metal-templated reaction accompanied by air oxidation. The complexes can exist indefinitely at ambient conditions without any sign of decomposition in water, nonaqueous solutions and in the solid state. We anticipate that our findings may open a way to aqueous solution and polynuclear high-valent iron chemistry that remains underexplored and presents an important challenge.

[1] Department of Chemistry, Taras Shevchenko National University of Kyiv, 64 Volodymyrska Street, 01601 Kiev, Ukraine. [2] Institute of Inorganic Chemistry and Analytical Chemistry, Johannes Gutenberg-University of Mainz, Duesbergweg 10-14, 55128 Mainz, Germany. [3] qLeap Center for Theoretical Chemistry, Department of Chemistry, University of Aarhus, DK-8000 Arhus C, Denmark. [4] Faculty of Chemistry, University of Wrocław, F. Joliot-Curie 14, 50-383 Wrocław, Poland. [5] Department of Inorganic Chemistry, Technische Universität Dresden, Bergstraße 66, D-01069 Dresden, Germany. [6] V.I. Vernadskii Institute of General and Inorganic Chemistry, National Academy of Sciences of Ukraine, 03680 Kiev, Ukraine. Correspondence and requests for materials should be addressed to I.O.F. (email: ifritsky@univ.kiev.ua).

High-valent iron ($+4$, $+5$, $+6$) does not occur in the mineral world as ferric ($+3$) species, the most stable natural form of iron in air and in water, does not react with atmospheric $O_2$ at ambient conditions. High-valent iron is believed to exist only as reactive transient species in the catalytic cycles of many haem and non-haem iron enzymes[1–4] as well as in important industrial (Haber–Bosch synthesis), laboratory and environmental (Fenton reaction) catalytic processes[4–7]. That is why the chemistry of high-valent has attracted enormous interest in recent years.

Within the past 20 years, tremendous progress has been achieved in this field, and many remarkable high-valent iron complexes belonging to various ligand families have been developed and successfully used not only as model compounds for mimicking enzymatic active sites but also as versatile catalysts[1–11]. One of the most spectacular examples of such compounds is a family of iron complexes based on a class of the tetraamide macrocyclic ligands (Fe-TAMLs) developed by Collins since the early 1990s (refs 12–17). Fe-TAML systems demonstrate not only remarkable stability and resistance to oxidative degradation but can also act as exceptionally efficient homogeneous catalysts in aqueous solutions exploiting high-valent iron oxo species[18,19]. They exhibit extraordinary ability to activate peroxides[9,19] for a variety of environmentally important catalytic oxidation reactions, in particular, oxidative degradation of organic azo dyes[20,21], removal of sulfur from hydrocarbon fuels[22] and destroying trace industrial and agricultural pollutants in water[9,11].

Although many reports dedicated to bioinspired oxo (Fe$^{IV,V}$=O), nitrido (Fe$^{IV,V}$≡N) and imido (Fe$^{IV}$=N–R) iron compounds have been published since 2000 (refs 8,12,13,23–33), new examples of non-biomimetic high-valent iron complexes are rare and remain much less explored[34–36]. The most recent examples of such compounds are astonishing iron(IV) salts of the decamethylferrocene dication reported by Malischewski et al.[37] a few months ago. New developments in this area are crucial not only for the elucidation of electronic structure and understanding the fundamental aspects of high-valent iron chemistry but also for perspectives on opening of new facets of this field, such as the use of these compounds for preparation of polynuclear assemblies and metal-organic frameworks as well as in molecular magnetism and materials science. The main things that limit and complicate the research and utility of high-valent iron compounds is low stability of most of them in polar organic solvents and especially in water, and a lack of facile synthetic protocols that would allow the preparation of these compounds in sufficient quantities with the use of the most convenient and accessible solvents and oxidants. Up to date, no stable high-valent iron compounds have been reported that could be easily synthesized with the use of both the nature's principal solvent (water) and the nature's principal oxidizing agent (atmospheric $O_2$), and at the same time would be comparable in stability with typical low-valent metal complexes.

Indeed, the vast majority of known high-valent iron compounds (both inorganic salts and coordination complexes) appear to be unstable in protic solvents, undergoing degradation in rather short time. Complexes with supporting organic ligands exhibit half-lives ($t_{1/2}$) in aqueous solution varying from parts of seconds to a few hours[7,12,23,28,38,39]. The reported iron(IV) nitride complexes indicate noticeable stability both in the solid state and in solution, they are air and moisture stable at ambient conditions[30] and do not react with water[29]. However, they are apparently not soluble in water and cannot be obtained in aqueous solutions, whereas the iron(V) nitrido complex reacts with water producing ammonia[29]. Ferrates(VI), considered for a long time as probably the most stable among known high-valent

iron compounds, decompose upon standing in aqueous solution within hours, evolving molecular oxygen[40]. Evidently, the most efficient systems providing extended lifetimes in aqueous media are high-valent Fe-TAMLs[12,14–16].

Most of the known high-valent iron complexes have been synthesized in organic solvents, and only a small subset has been obtained in aqueous media from low-valent precursors, with the use of various oxidants (for example, ammonium cerium(IV) nitrate, peroxides)[12,14,23] but not atmospheric $O_2$. An aqua ferrous ($+2$) species can react with ozone in acidic aqueous solution producing highly reactive oxoiron(IV) species that, however, decays within seconds[7]. Many other ferrous species can be air oxidized to $+3$ state in water but, as we mentioned above, all known ferric species do not react with atmospheric $O_2$ in aqueous media. In particular, Fe-TAMLs are stable as ferric species in aqueous solution and cannot be oxidized by oxygen in pure water[10] but only in weakly coordinating organic solvents such as methylene chloride[15]. Only recently, Collins and colleagues[10,41] have shown that direct dioxygen activation by Fe$^{III}$-TAMLs can be achieved in reverse micelles of aqueous aerosol OT in n-octane. However, the oxidized iron complexes were not isolated in the latter case, and the high-valent species were identified only as reactive intermediates in catalytic oxidation reactions[10,41].

Herein, we report unprecedented water-soluble iron(IV) clathrochelate complexes that can exist indefinitely long at ambient conditions without any sign of destruction, both in the solid state and solution. Moreover, these are also the first examples of stable high-valent iron compounds formed spontaneously as a result of air oxidation of low-valent iron species in water.

## Results

**Reaction design**. In search of ligand systems that provide an extraordinarily efficient stabilization of high-valent iron, we have been inspired by TAML-based complexes, as they are among the most stable high-valent iron species reported up to date (in particular, in aqueous solution)[12–21]. The deprotonated amide groups are known to be one of the best donors for the stabilization of high oxidation states of transitional metals. Particular efficacy of TAMLs is because of both strong σ-donor capacity and high total negative charge that can provide fully deprotonated polydentate ligands. Another impetus was our recent finding that the deprotonated hydrazide groups of the tetradentate macrocyclic and open-chain ligands are significantly more efficient donors for the stabilization of copper(III) than the amide groups[42]. As in most of its compounds iron exhibits a hexacoordinate geometry, we have considered hypothetical hexadentate hydrazide-containing ligands as particularly promising for an efficient stabilization of high-valent iron. In order to construct such a ligand, we have applied the methodology of template synthesis aiming to obtain clathrochelates, or cage metal complexes[43]. These have a remarkable ability to shield a metal ion from external factors (for example, effects of solvation, ligand substitution) by encapsulating it within a three-dimensional macropolycyclic ligand cavity that makes them attractive for stabilizing unusual oxidation states of transition metal ions[43,44].

**Synthesis and characterization**. One-pot template reaction between iron(III) nitrate, oxalodihydrazide (**oxh**) and formaldehyde in the presence of atmospheric $O_2$ proceeds smoothly in alkaline aqueous media resulting in the anionic clathrochelate complex [Fe$^{IV}$(**L**-6H)]$^{2-}$, where **L** is the macrobicyclic hexahydrazide ligand constructed by linking three **oxh** residues with six

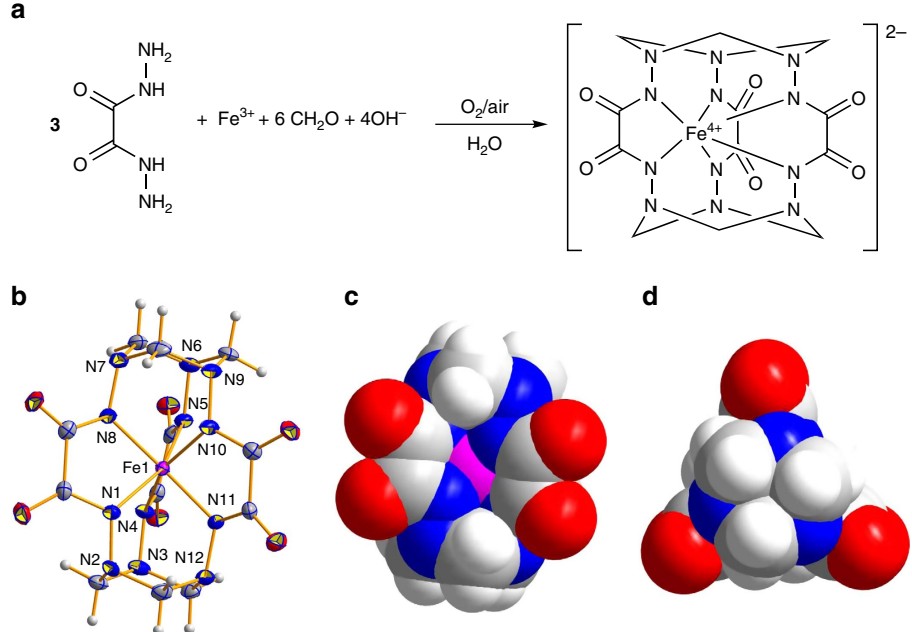

**Figure 1 | Synthesis and X-ray crystal structure of hexahydrazide cage iron(IV) complexes.** (**a**) General synthetic scheme. (**b**) Molecular structure of the complex anion $[Fe^{IV}(\textbf{L}\text{-}6H)]^{2-}$ in **3** showing the atomic numbering scheme (displacement ellipsoids are drawn at the 50% probability level). Carbon atoms are shown in dark grey, oxygen atoms in red, nitrogen in blue, iron in magenta and hydrogen in light grey. (**c**) Space-filling representation of the complex anion $[Fe^{IV}(\textbf{L}\text{-}6H)]^{2-}$ in **3** (side view). (**d**) Space-filling representation of the complex anion (top view). Selected bond lengths (Å) and angles (°): Fe1–N1 1.968(2), Fe1–N4 1.952(3), Fe1–N5 1.969(3), Fe1–N8 1.945(3), Fe1–N10 1.950(3), Fe1–N11 1.958(3), N1–Fe1–N8 79.46(12), N4–Fe1–N5 78.84(12), N10–Fe1–N11 79.39(12), N1–Fe1–N4 86.76(12), N8–Fe1–N10 86.87(13), N1–Fe1–N11 86.65(12).

methylene bridges (Fig. 1a). The formation of the cage complex is manifested by the development of a deep green colour of the reaction mixture. The negative-mode electrospray ionization high-resolution mass spectrum (ESI-HRMS) of the reaction solution exhibits two prominent peaks at a mass-to-charge ratio $m/z = 477.0463$ and $499.0279$, corresponding to $\{[Fe^{IV}(\textbf{L}\text{-}6H)]^{2-} + H^+\}^-$ and $\{[Fe^{IV}(\textbf{L}\text{-}6H)]^{2-} + Na^+\}^-$ species, respectively (Supplementary Fig. 1). When the reaction is carried out under a nitrogen atmosphere, the mixture has a brown colour that becomes dark green upon exposure to air.

With the use of different alkaline reagents ($NH_3 \cdot H_2O$, $NBu_4OH$, $Ca(OH)_2$) we succeeded in isolating three anionic complexes $(C_6N_4H_{13})_2[Fe^{IV}(\textbf{L}\text{-}6H)] \cdot 5H_2O$ (**1**, containing hexamethylenetetraminium cation formed by the reaction of formaldehyde with aqueous ammonia), $(Bu_4N)_2[Fe^{IV}(\textbf{L}\text{-}6H)] \cdot 7CHCl_3$ (**2**), $Ca(H_2O)_2Fe^{IV}(\textbf{L}\text{-}6H) \cdot 4H_2O \cdot i\text{-}PrOH$ (**5**), respectively, as dark green crystalline materials. Two other complexes were prepared by the metathesis of $C_6N_4H_{13}^+$ cations of **1** with $Ph_4AsBr$ in aqueous solution: $(Ph_4As)_2[Fe^{IV}(\textbf{L}\text{-}6H)] \cdot 13.28H_2O$ (**3**), or $Bu_4N^+$ cations of **2** with $NaClO_4$ in acetone: $Na_2[Fe^{IV}(\textbf{L}\text{-}6H)] \cdot 2H_2O$ (**4**). All of the obtained complexes are readily soluble in water, whereas those with organic cations are also soluble in many polar organic solvents. The intense green colour of their solutions is attributed to strong absorption in the visible region with distinct maxima at 651 nm ($\varepsilon = 8,900\ M^{-1}\ cm^{-1}$) and 422 nm ($6,700\ M^{-1}\ cm^{-1}$) (Fig. 2).

The isolated iron(IV) clathrochelates are surprisingly stable in both aqueous/nonaqueous solutions and the solid state. Ultraviolet–visible (UV–Vis) spectral monitoring of $10^{-4}\ M$ aqueous solution of **3** at pH 7.0 demonstrated the absence of any spectral decrease over a 6-month period, and at pH 1.0 and 13.0 the summary intensity decay was < 3% over the course of 30 days (Supplementary Figs 2–4). The Mössbauer spectrum of **3**, collected 1 year after the isolation of the complex, revealed no changes (Table 1 and Supplementary Fig. 5). To the best of our

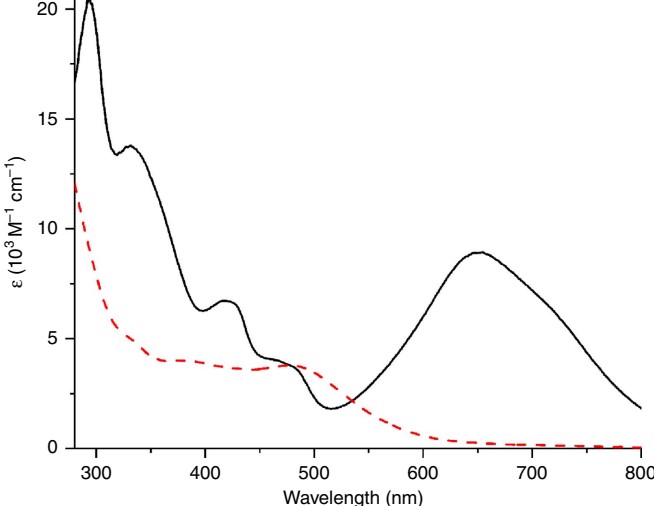

**Figure 2 | Electronic absorption spectra of 3 recorded at 20 °C.** The $10^{-4}\ M$ aqueous solution (solid line) and of $10^{-4}\ M$ aqueous solution upon addition of 500 equiv. of sodium dithionite (1.5 M aqueous solution) at pH = 10.01 (dashed line).

knowledge, this is the first example of a high-valent iron compound formed in aqueous media in the presence of atmospheric $O_2$ that is indefinitely stable under those conditions.

**Crystallographic study.** X-ray single-crystal analysis of **2**, **3** and **5** revealed the presence of the clathrochelate complex dianions $[Fe^{IV}(\textbf{L}\text{-}6H)]^{2-}$ containing the encapsulated $Fe^{4+}$ ion coordinated by six deprotonated hydrazide nitrogen atoms (Fig. 1b,c). The macropolycyclic ligand can be described as the

**Table 1 | Experimental and DFT(BP86)-calculated Mössbauer parameters of the synthesized complexes.**

| Complex | T, K | $\delta$, mm s$^{-1}$ | $\Delta E_Q$, mm s$^{-1}$ | $\Gamma_{FWHM}$, mm s$^{-1}$ |
|---|---|---|---|---|
| **1** | 80 | 0.116 (2) | 2.495 (4) | 0.170 (3) |
| **1** | 293 | 0.037 (6) | 2.471 (10) | 0.156 (8) |
| **3** | 80 | 0.121 (3) | 2.505 (5) | 0.149 (4) |
| **3** | 293 | 0.045 (5) | 2.511 (10) | 0.151 (8) |
| **3** | DFT | 0.077 | − 2.398 | |
| **3** | DFT (OPT) | 0.078 | − 2.503 | |
| **3**\* | 293 | 0.044 (3) | 2.499 (6) | 0.134 (4) |
| **3**† | 80 | 0.12 (1) | 2.43 (2) | 0.13 (1) |
| **3**‡ | 80 | 0.25 (3) | 1.12 (6) | 0.29 (4) |

DFT, density functional theory.
OPT denotes calculation using DFT-optimized structure.
\*Recorded 1 year after isolation of the complex.
†Frozen aqueous solution.
‡Frozen aqueous solution reduced in the presence of excess of $Na_2S_2O_4$.

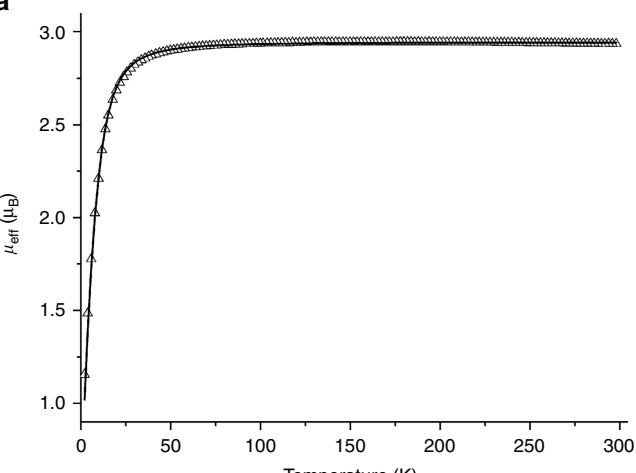

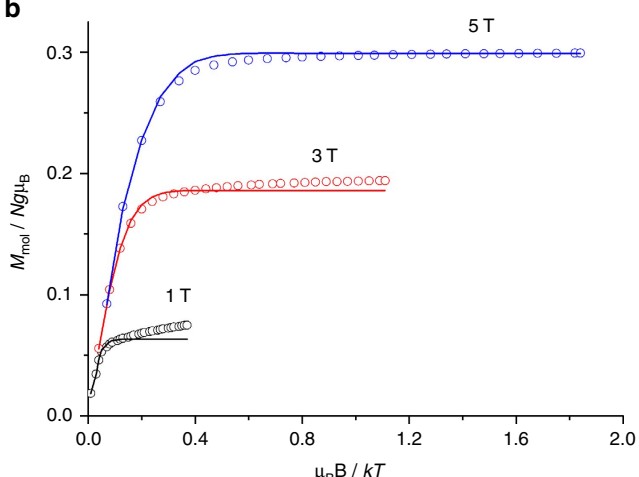

**Figure 3 | Magnetochemical characterization of 3.** (**a**) Temperature dependence of magnetic moment for the crystalline sample of **3**. The solid line represents the best fit obtained with the parameters given in the text. (**b**) Field-dependent magnetization measurements at variable temperature (as $M_{mol}$ versus $B/T$) for the crystalline sample of **3**. Simulation of the experimental data with the use of the spin Hamiltonian and taking into account zero-field and Zeeman splitting (equation (1)) resulted in the following parameters values: $g = 2.05$ and $D = + 20.7$ cm$^{-1}$.

dodeca-aza-quadricyclic cage framework with two capping 1,3,5-triazacyclohexane fragments consisting of three five- and six six-membered alternating chelate rings. The coordination geometry of the metal centres is intermediate between a trigonal prism (distortion angle $\varphi = 0°$) and a trigonal antiprism ($\varphi = 60°$) with $\varphi$ in the range 28.0°–33.1° (Supplementary Table 1). The Fe–N bond distances in **2**, **3** and **5** are in the range 1.915(5)–1.969(3) Å (Supplementary Table 2) that are somewhat longer than those reported for the iron(IV)-TAML complexes (1.88–1.93 Å)[14,15]. Although the structures of **2** and **3** are ionic and contain the isolated counter cations ($Bu_4N^+$ in **2**, $Ph_4As^+$ in **3**), **5** is a coordination polymer in which $Ca^{2+}$ cations are exo-coordinated to the vacant (O,O') and (O,N) chelating units, thus uniting three neighbouring complex anions in two-dimensional networks (Supplementary Figs 6 and 7).

**Magnetochemical and Mössbauer spectroscopy studies.** Complex **3** was subjected to a SQUID (superconducting quantum interference device) magnetic susceptibility measurement in the temperature range 2–300 K. The value of the effective magnetic moment, $\mu_{eff}$, at 293 K of 2.95 $\mu_B$ (Fig. 3a) is close to the spin-only value for two unpaired electrons (2.83 $\mu_B$) and clearly shows the intermediate spin state of iron(IV) in this complex[25,35,45,46]. Above 40 K, the value of $\mu_{eff}$ is nearly temperature independent, whereas below this temperature $\mu_{eff}$ decreases steeply because of zero-field splitting. The experimental data were fitted with the use of the spin Hamiltonian and taking into account zero-field and Zeeman splitting:

$$\hat{H} = D\left[\hat{S}_z^2 - \frac{1}{3}S(S+1)\right] + g\mu_B\vec{B}\cdot\vec{S} \quad (1)$$

Simulations fit well with $g = 2.08$ and $D = + 23.1$ cm$^{-1}$ (the sign of $D$ was determined from the isofield magnetization measurements at variable temperatures (Fig. 3b), and the obtained $D$ value is close to those reported for the known $S = 1$ iron(IV) complexes[8,25,46].

The zero-field $^{57}$Fe Mössbauer spectra of **1** and **3** collected at 80 and 293 K confirmed assignment of the oxidation state: the spectrum of **3** at 80 K displays a quadrupole doublet with an isomer shift $\delta = 0.121(3)$ mm s$^{-1}$ and a quadrupole splitting $|\Delta E_Q| = 2.505(5)$ mm s$^{-1}$ (Fig. 4 and Table 1). These parameters are consistent with the values expected for the intermediate spin d$^4$ complexes[5,8,46]. Large $|\Delta E_Q|$ indicates relatively enhanced value of the electric field gradient arising from the trigonal distortion. The observed $\delta$ values fall within the middle of the range reported for other iron(IV) compounds[5,8,29,30,37,46] but appreciably greater than those reported for the iron(IV)

complexes with TAML ligands ($-0.04$ to $-0.19$ mm s$^{-1}$)[12,14]. The more positive isomer shift for **1** and **3** indicates a diminishing s-electron density on iron nuclei and the less covalent Fe–N bonding as compared with the iron(IV)-TAML complexes, and correlates to longer Fe–N bond lengths in the iron(IV) hexahydrazide complexes.

**Quantum chemical calculation.** Density functional theory (DFT) calculations (BP86/TZVP method) of **3** corroborate the assumed electronic structure: an iron(IV) intermediate (triplet) spin state was found to be the most energetically favourable. It is separated significantly from the hypothetical low- and high-spin iron(IV) prismatic complexes by 22 and 28 kcal mol$^{-1}$, respectively. The DFT-optimized geometrical structure parameters are in an excellent agreement with the experimental results (Fig. 5a and Supplementary Table 3).

The molecular orbital structure within the chosen coordinate system reveals DOMO (doubly occupied molecular orbital) to be almost pure Fe-derived $d_{z^2}$ orbital (Fig. 5b). Both $d_{x^2-y^2-}$ and $d_{xy-}$ based degenerate SOMOs (singly occupied molecular

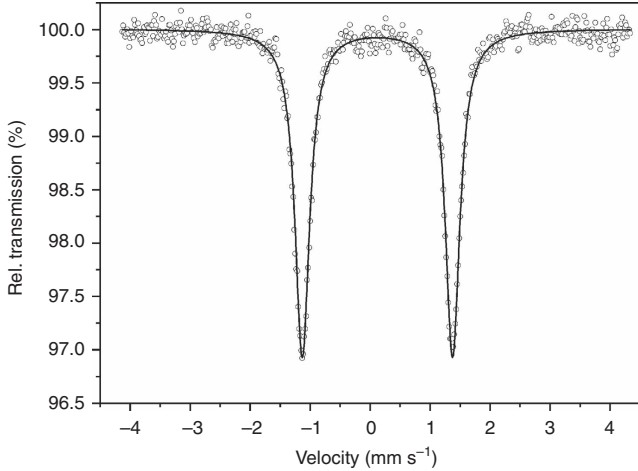

**Figure 4 | Zero-field $^{57}$Fe Mössbauer spectrum of a microcrystalline sample of 3 recorded at 80 K.** The solid lines represent the calculated Lorentzian doublet with the parameters given in Table 1.

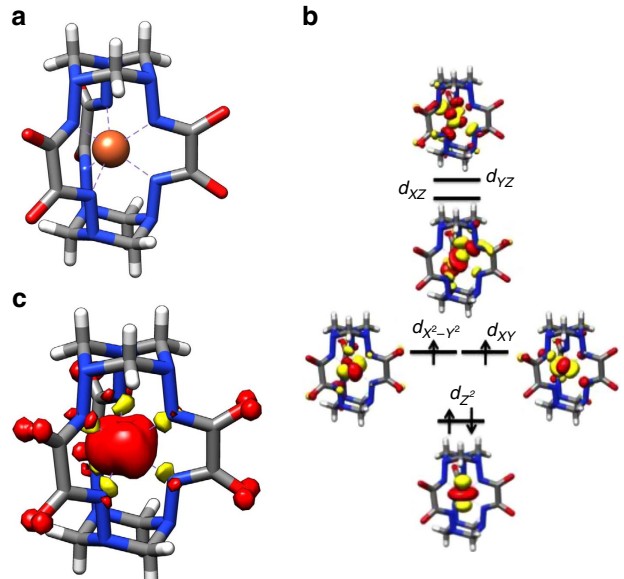

**Figure 5 | DFT electronic structure calculations. (a)** Computational model (initially derived from the complex anion of **3**) consists of 43 atoms and has 1,087 contracted basis functions. The colour scheme: red, oxygen; blue, nitrogen; grey, carbon; white, hydrogen; and orange, iron. **(b)** The frontier molecular orbitals diagram for complex anion of **3**. **(c)** Spin density for complex anion of **3**.

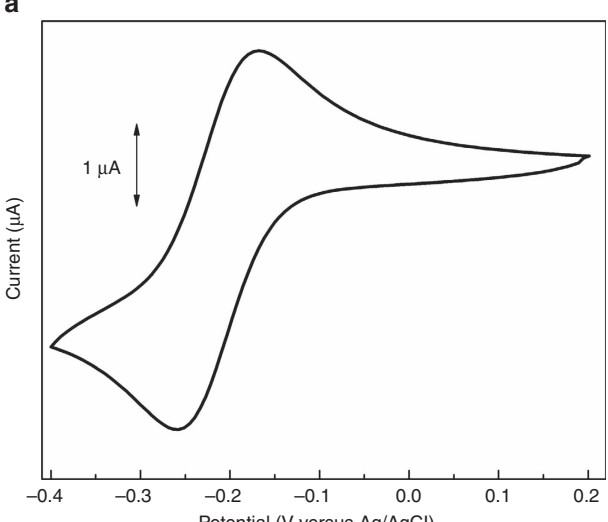

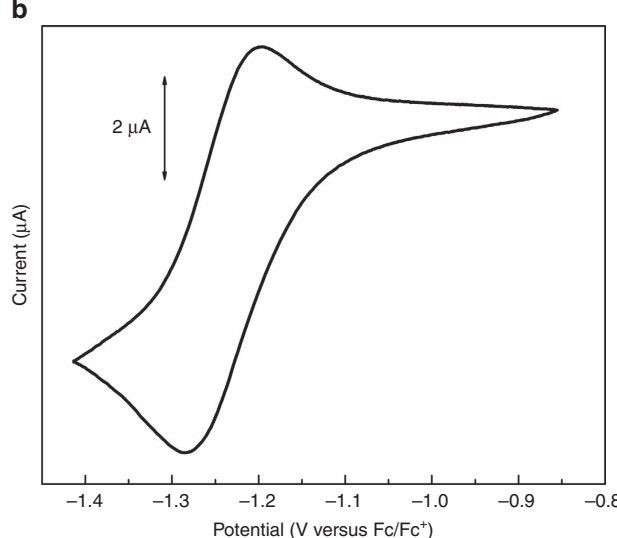

**Figure 6 | The cyclic voltammograms of 3 at a scan rate of 25 mV s$^{-1}$.** (**a**) In aqueous solution (1 mM) with NaClO$_4$ (0.1 M) as supporting electrolyte. (**b**) In acetonitrile solution (1 mM) with Bu$_4$NClO$_4$ (0.1 M) as supporting electrolyte. For the numerical data and cyclic voltammograms recorded at different scan rates, see Supplementary Tables 4,5 and Supplementary Figs 8,10.

hydrazido groups into the $d_{xz}$ and $d_{yz}$ orbitals (the electrons in $d_{z^2}$, $d_{xz}$ and $d_{yz}$ orbitals yield a negative contribution to $\Delta E_Q$ (refs 12,16,17).

**Cyclic voltammtery study.** The cyclic voltammograms of **3** reveal quasireversible features at $E_{1/2} = -0.21$ V versus Ag/AgCl with $\Delta E_p = 76$ mV in aqueous and at $E_{1/2} = -1.24$ V versus Fc/Fc$^+$ with $\Delta E_p = 66$ mV in acetonitrile solutions because of the metal-centred Fe$^{4+/3+}$ redox process (Fig. 6, Supplementary Figs 8–10 Supplementary Tables 4 and 5). These values are substantially more negative than those reported for the known iron(IV) complexes and can be compared with the most negative potentials observed for the Fe$^{+4/+3}$ couple in acetonitrile (ca. $-1.0$ V)$^{26,46,48}$. These data are indicative of very efficient thermodynamic stabilization of the Fe$^{4+}$ oxidation state in the cage compounds brought about by strong σ-donor capacity and

orbitals) are essentially nonbonding orbitals with very strong metal character. The LUMO (lowest unoccupied molecular orbital) $d_{xz}$ and LUMO + 1 $d_{yz}$ antibonding orbitals are much higher in energy. The spin density (Fig. 5c) almost exclusively localized on the metal centre (1.89 according to Mulliken population analysis).

Both the calculated δ (0.078 mm s$^{-1}$) and $\Delta E_Q$ ($-2.503$ mm s$^{-1}$) values also agree well with the experimental Mössbauer parameters (Table 1), thus providing additional strong evidence for the correct assignment of the electronic ground state. As the valence contribution to the electric field gradient (EFG) for 3d configuration $(z^2)^2(x^2-y^2)^1(xy)^1$ is zero$^{37,47}$, the large negative $\Delta E_Q$ can be explained by noticeable negative covalence contribution because of electron density donation from the

high total negative charge of the fully deprotonated hexahydrazide encapsulating ligand. In addition, in acenonitrile solution **3** shows a quasireversible wave at $E_{1/2} = 0.03\,V$ versus Fc/Fc$^+$ with $\Delta E_p = 72\,mV$ that most probably corresponds to the Fe$^{5+/4+}$ couple (Supplementary Figs 9 and 11 and Supplementary Table 5).

**Reduction to iron(III) and reactivity**. The iron(IV) complexes **1–5** are poor oxidants reacting only with relatively strong and moderate reductants (such as sodium dithionite, sulfide, ascorbate) in aqueous solution being turned to the corresponding iron(III) clathrochelates that is manifested by a distinctive colour change from deep green to brown. The UV–Vis spectra of the obtained solutions are identical to those of iron(III) complexes formed under nitrogen atmosphere (*vide supra*). The $^{57}$Fe Mössbauer spectrum of frozen aqueous solution of **3** reduced in the presence of excess of sodium dithionite recorded at 80 K exhibits a quadrupole doublet (Supplementary Fig. 12 and Table 1) that differs distinctly from that of **3** indicating significantly more positive $\delta = 0.25(3)$ mm s$^{-1}$ ($\Delta\delta = 0.13$ mm s$^{-1}$), indicating a metal-centred reduction resulting in the formation of the low-spin ($S = 1/2$) ferric species[12,14,47,49]. Smaller $|\Delta E_Q|$ value (1.12 (6) mm s$^{-1}$) compared with that in **3** (2.505 (5) mm s$^{-1}$) can be explained by the appearance of a positive valence contribution to the EFG and thus to $\Delta E_Q$ caused by adding one electron into the $d_{x^2-y^2}$, $d_{xy}$ level[12,16,17,47]. The negative terms of covalence contribution to the EFG are approximately equal for iron(IV) and iron(III) complexes (as in both corresponding three-dimensional configurations the $d_{xz}$ and $d_{yz}$ orbitals, giving rise to the negative covalence contribution, are formally empty). The estimated resulting $|\Delta E_Q|$ in the iron(III) complex is then a sum of the positive valence and the negative covalence contributions. Therefore, the observed $|\Delta E_Q|$ is more positive than in **3** but at the same time it is somewhat smaller than expected because of the fact that the total ligand contributions are in reality not exactly equal (more negative in the case of the iron(III) species).

Interestingly, the reduced solutions containing iron(III) species appeared to be EPR silent at both room and liquid nitrogen temperatures that is not surprising for many iron(III) complexes. Evidently, registration of the EPR signal for these low-spin iron(III) complexes with an unusual configuration $d_{z^2}^2 d_{xy}^2 d_{x^2-y^2}^1$ or $d_{z^2}^2 d_{xy}^1 d_{x^2-y^2}^2$ awaits future low-temperature experiments at liquid helium temperatures.

In contrast to all known ferric species, the iron(III) cage complexes are oxidized by oxygen in water: their brown aqueous solutions, when exposed to air, quickly recover the initial green colour indicating the regeneration of iron(IV) by atmospheric oxidation (Supplementary Fig. 13). This is the first example of ferric species that are not stable at ambient aqueous conditions and undergo spontaneous air oxidation. Remarkably, this proceeds without observable loss of initial extinction, even at low concentrations of the complexes and high excess of reducing agents, as shown for $10^{-4}$ M aqueous solutions of **3** reduced by 500 equiv. of sodium dithionite.

## Discussion

In conclusion, we report a new class of high-valent iron compounds, formed by atmospheric oxidation in aqueous media, that are extremely stable in both the solid and solution states, and that can exist indefinitely under ambient conditions, exhibiting no signs of degradation. The exceptionally efficient stabilization of the iron(IV) oxidation state can be attributed to the highly (6−) negative charge of the clathrochelate ligand combined with a strong electron-donor effect from the six deprotonated

hydrazide nitrogen atoms, as well as a shielding effect provided by the macropolycyclic cage on the metal ion. The observed enormous aqueous stability clearly distinguishes the presented iron(IV) complexes from most of the earlier reported high-valent iron compounds that are typically unstable, highly reactive and quickly decomposing species in aqueous media. To the best of our knowledge, these are the first high-valent iron complexes indefinitely stable in water. Moreover, they are stable at least for weeks under aggressive (strongly acidic or basic) conditions.

The results show that novel iron(IV) complexes are readily accessible from simple, commercially available, inexpensive starting materials under surprisingly mild reaction conditions. The preparative methods leading to the hexahydrazide cage iron(IV) complexes are quite simple, reproducible and require no special techniques (for example, Schlenk line or glove box) or low temperatures. The remarkable template reactions can be easily carried out in test tubes, even in minimally equipped laboratories. The robust nature of the cage complexes, solubility in various solvents, and notable reversibility of Fe$^{4+/3+}$ reduction/oxidation showcases the potential of their application in redox catalysis, and the presence of vacant chelate ribbed units makes them promising building blocks for polynuclear assemblies.

## Methods

**Materials**. All the reagents and solvents used in this work were purchased from commercial sources and were used as received without further purification. Ligand **oxh** (99.99%, Aldrich) was used without further purification. Dry acetonitrile (CH$_3$CN) was degassed and purified under nitrogen atmosphere.

**Synthesis of (C$_6$N$_4$H$_{13}$)$_2$[Fe(L-6H)]·5H$_2$O (1)**. Freshly prepared iron(III) hydroxide (obtained by mixing of Fe(NO$_3$)$_3$·9H$_2$O (0.133 g, 0.33 mmol, dissolved in 5 ml of water) and aqueous ammonia (28%, 0.17 ml, 1.67 mmol) with consequent filtering and washing with water (15 ml) of the formed precipitate) was added to the warm (∼50 °C) solution of oxalodihydrazide (0.120 g, 1 mmol) in 10 ml of water upon stirring. Then, excess quantities of aqueous ammonia (28%, 0.52 ml, 5 mmol) and an aqueous formaldehyde solution (37% in water, 0.5 ml, 6.7 mmol) were added immediately. The resulting mixture was stained a dark green colour within few minutes. Then, the reaction mixture was stirred for 30 min at the ambient temperature, filtered off and the solvent was removed on a rotary evaporator. The resulting dark green powder was washed with chloroform (20 ml), air dried and then treated with 30 ml of cold (∼ −10 °C) 70% aqueous ethanol. The obtained solution was filtered and a small amount of brown insoluble material was discarded. The filtrate was reduced in volume till ca. 3 ml on rotary evaporator and then set aside in an open vessel at room temperature. In 24 h, the formed precipitate was filtered, washed with cold water and air dried. Additional amount of the product can be obtained as a result of further evaporation of the mother liquor in air. The results of elemental analysis showed that the obtained product is a complex with hexamethylenetetraminium cation formed by the reaction of formaldehyde with aqueous ammonia[50]. Yield 0.19 g (67%). Elemental analysis for C$_{24}$H$_{48}$N$_{20}$O$_{11}$Fe (848.62): calculated, %: C, 33.97; H, 5.70; N, 33.01. Found, %: C, 33.71; H, 5.86; N, 33.10. ESI-HRMS (*m/z*): [M + H$^+$]$^-$ calcd for C$_{12}$H$_{13}$N$_{12}$O$_6$Fe 477.0436. Found: 477.0460. [M + Na$^+$]$^-$ calcd for C$_{12}$H$_{12}$N$_{12}$O$_6$FeNa 499.0255. Found: 499.0277. Infrared absorption bands (w, weak, m, medium, s, strong, versus, very strong): 3,418 cm$^{-1}$ (s) O–H stretch, 3,190 cm$^{-1}$ (s) O–H stretch, 3,019 cm$^{-1}$ (w) C–H stretch, 2,957 cm$^{-1}$ (w) C–H stretch, 1,632 cm$^{-1}$ (vs) C = O stretch, Amide I, 1,440 cm$^{-1}$ (m), 1,401 cm$^{-1}$ (m), 1,374 cm$^{-1}$ (m), 1,264 cm$^{-1}$ (m), 1,184 cm$^{-1}$ (w), 1,155 cm$^{-1}$ (w), 1,099 cm$^{-1}$ (w), 1,041 cm$^{-1}$ (m), 1,007 cm$^{-1}$ (w), 980 cm$^{-1}$ (w), 951 cm$^{-1}$ (w), 916 cm$^{-1}$ (w), 857 cm$^{-1}$ (w), 822 cm$^{-1}$ (w), 790 cm$^{-1}$ (w), 756 cm$^{-1}$ (w), 738 cm$^{-1}$ (w), 632 cm$^{-1}$ (w), 615 cm$^{-1}$ (w), 667 cm$^{-1}$ (m), 510 cm$^{-1}$ (w), 460 cm$^{-1}$ (w), 416 cm$^{-1}$ (w).

**Synthesis of (Bu$_4$N)$_2$[Fe(L-6H)]·7CHCl$_3$ (2)**. FeCl$_3$·6H$_2$O (0.270 g, 1 mmol, dissolved in 5 ml of water) was added to the warm (∼50 °C) solution of oxalo-dihydrazide (0.361 g, 3 mmol) in 10 ml of water. Then, tetrabutylammonium hydroxide (40% aqueous solution, 3.26 ml, 5 mmol) and an aqueous formaldehyde solution (7% in water, 0.68 ml, 9 mmol) were added immediately to the resulting mixture. The reaction mixture was stirred for 30 min at room temperature, filtered off and the filtrate was removed on a rotary evaporator. The resulting residue was extracted by 50 ml of chloroform, filtered off, the filtrate was dried over anhydrous Na$_2$SO$_4$, filtered off and then filtered through a M60 silica gel layer. After that the filtrate was concentrated *in vacuo* till volume of ca. 10 ml. Dark black single crystals were obtained in 48 h by slow diffusion of hexane vapours into the filtrate, picked from solution, soaked with filter paper and air dried. After drying in vacuum, the

crystals lose the solvate chloroform, and the elemental analysis data correspond to the composition **(Bu$_4$N)$_2$[Fe(L-6H)]**. Yield 0.82 g (85%). Elemental analysis for C$_{44}$H$_{84}$N$_{14}$O$_6$Fe (961.09): calculated, %: C, 54.99; H, 8.81; N, 20.40. Found, %: C, 55.07; H, 8.66; N, 20.19. ESI-HRMS (m/z): [M + H$^+$]$^-$ calcd for C$_{12}$H$_{13}$N$_{12}$O$_6$Fe 477.0436. Found: 477.0460. [M + Na$^+$]$^-$ calcd for C$_{12}$H$_{12}$N$_{12}$O$_6$FeNa 499.0255. Found: 499.0277.

**Synthesis of (Ph$_4$As)$_2$[Fe(L-6H)] · 13.28H$_2$O (3).** **1** (0.085 g, 0.1 mmol) was dissolved in water (4 ml) and then Ph$_4$AsBr (0.084 g, 0.18 mmol, dissolved in 4 ml of water) was added. The volume of the resulting solution was reduced on a rotary evaporator till ca. 3 ml, and the green product was extracted by chloroform (3 × 10 ml). The combined chloroform extracts were dried over anhydrous Na$_2$SO$_4$, filtered off, and after that the solvent was completely removed. The residue was dissolved in a small amount (5 ml) of warm water and set aside for crystallization at the ambient conditions. Single crystals suitable for X-ray analysis were obtained in 48 h (Supplementary Fig. 14). Yield 0.11 g (83%). Elemental analysis for C$_{60}$H$_{78.56}$N$_{12}$O$_{19.28}$FeAs$_2$ (1482.12): calculated, %: C, 48.62; H, 5.34; N, 11.34. Found, %: C, 48.82; H, 5.36; N, 11.25. ESI-HRMS (m/z): [M + H$^+$]$^-$ calcd for C$_{12}$H$_{13}$N$_{12}$O$_6$Fe 477.0436. Found: 477.0463. [M + Na$^+$]$^-$ calcd for C$_{12}$H$_{12}$N$_{12}$O$_6$FeNa 499.0255. Found: 499.0279. Infrared absorption bands: 3,418 cm$^{-1}$(s) O–H stretch, 3,054 cm$^{-1}$(w) C–H stretch, 2,943 cm$^{-1}$(w) C–H stretch, 1,637 cm$^{-1}$(vs) C = O stretch, Amide I, 1,481 cm$^{-1}$(w), 1,440 cm$^{-1}$(m), 1,355 cm$^{-1}$(m), 1,336 cm$^{-1}$(m), 1,198 cm$^{-1}$(w), 1,168 cm$^{-1}$(w), 1,080 cm$^{-1}$(w), 1,024 cm$^{-1}$(w), 993 cm$^{-1}$(m), 945 cm$^{-1}$(w), 916 cm$^{-1}$(m), 895 cm$^{-1}$(w), 745 cm$^{-1}$(s), 690 cm$^{-1}$(m), 660 cm$^{-1}$(m), 634 cm$^{-1}$(m), 523 cm$^{-1}$(w), 478 cm$^{-1}$(w), 468 cm$^{-1}$(m), 446 cm$^{-1}$(w), 410 cm$^{-1}$(w). UV–vis (H$_2$O): λ$_{max}$ (ε, mol$^{-1}$ dm$^3$ cm$^{-1}$) 294 nm (20,500), 332 nm (13,800), 418 nm (6,700), 475 nm (shoulder, 3,850), 651 nm (8,900).

**Synthesis of Na$_2$[Fe(L-6H)] · 2H$_2$O (4).** **2** (0.096 g, 0.1 mmol) was dissolved in 3 ml of acetone and overflow amount of NaClO$_4$ · H$_2$O (0.042 g, 0.3 mmol) dissolved in 2 ml of acetone was added. The green precipitate was filtered off, washed with acetone and dried in the air. Yield 0.050 g (96%). Elemental analysis for C$_{12}$H$_{16}$N$_{12}$O$_8$Na$_2$Fe (558.16): calculated, %: C, 25.82; H, 2.89; N, 30.11. Found, %: C, 26.03; H, 3.06; N, 29.97. ESI-HRMS (m/z): [M + H$^+$]$^-$ calcd for C$_{12}$H$_{13}$N$_{12}$O$_6$Fe 477.0436. Found: 477.0463. [M + Na$^+$]$^-$ calcd for C$_{12}$H$_{12}$N$_{12}$O$_6$FeNa 499.0255. Found: 499.0279. Infrared absorption bands: 3,418 cm$^{-1}$(s) O–H stretch, 2,952 cm$^{-1}$(m) C–H stretch, 1,642 cm$^{-1}$(vs) C = O stretch, Amide I, 1,612 cm$^{-1}$(s), 1,431 cm$^{-1}$(m), 1,392 cm$^{-1}$(m), 1,385 cm$^{-1}$(m), 1,300 cm$^{-1}$(m), 1,202 cm$^{-1}$(m), 1,175 cm$^{-1}$(w), 1,112 cm$^{-1}$(m), 1,095 cm$^{-1}$(m), 1,030 cm$^{-1}$(m), 996 cm$^{-1}$(w), 948 cm$^{-1}$(w), 918 cm$^{-1}$(w), 897 cm$^{-1}$(m), 745 cm$^{-1}$(w), 659 cm$^{-1}$(w), 638 cm$^{-1}$(w), 455 cm$^{-1}$(w), 437 cm$^{-1}$(w).

**Synthesis of Ca(H$_2$O)$_2$Fe(L-6H) · 4H$_2$O · i-PrOH (5).** FeCl$_3$ · 6H$_2$O (0.270 g, 1 mmol, dissolved in 5 ml of water) was added to the warm (∼50 °C) solution of oxalodihydrazide (0.361 g, 3 mmol) in 10 ml of water. Then, a hot (∼90 °C) mixture of an aqueous formaldehyde solution (37% in water, 0.68 ml, 9 mmol) and calcium hydroxide (0.185 g, 2.5 mmol) was added immediately to the resulting mixture. The reaction mixture was stirred for 30 min at the ambient temperature, filtered off and set aside for the solvent evaporation. The resulting powder was dissolved in 20 ml of warm (∼50 °C) water and filtered off. Dark green single crystals suitable for the X-ray analysis were obtained in 7 days by slow diffusion of isopropanol vapour into the filtrate, isolated by filtration, washed with isopropanol and air dried. Yield 0.46 g (0.67%). Elemental analysis for C$_{15}$H$_{32}$N$_{12}$O$_{13}$CaFe (684.42): calculated, %: C, 26.32; H, 4.71; N, 24.56. Found, %: C, 26.48; H, 4.58; N, 24.39. ESI-HRMS (m/z): [M + H$^+$]$^-$ calcd for C$_{12}$H$_{13}$N$_{12}$O$_6$Fe 477.0436. Found: 477.0461. [M + Na$^+$]$^-$ calcd for C$_{12}$H$_{12}$N$_{12}$O$_6$FeNa 499.0255. Found: 499.0277. Infrared absorption bands: 3,431 cm$^{-1}$(vs) O–H stretch, 3,251 cm$^{-1}$(s) O–H stretch, 2,936 cm$^{-1}$(m) C–H stretch, 1,724 cm$^{-1}$(s), 1,678 cm$^{-1}$(vs), 1,505 cm$^{-1}$(m), 1,422 cm$^{-1}$(m), 1,367 cm$^{-1}$(m), 1,316 cm$^{-1}$(m), 1,182 cm$^{-1}$(m), 1,112 cm$^{-1}$(m), 1,015 cm$^{-1}$(m), 876 cm$^{-1}$(m), 797 cm$^{-1}$(s), 751 cm$^{-1}$(m), 545 cm$^{-1}$(m).

**Elemental analysis.** Elemental analysis was conducted by the Microanalytical Service of the University of Kyiv.

**Mass-spectrometry measurements.** ESI-HRMS were collected on a Finnigan TSQ 700 mass spectrometer. Complexes were dissolved in water or in water/methanol (1:9) mixture to obtain solutions with concentrations of 10$^{-4}$–10$^{-6}$ M.

**Infrared vibrational spectroscopic measurements.** Infrared spectra were recorded on a Perkin-Elmer 180 spectrometer in the range of 400–4,000 cm$^{-1}$. Solid samples of the compounds were homogenized with excess amounts of KBr and a pressed pellet was measured at room temperature.

**Electronic absorption (UV–VIS) spectroscopic measurements.** Electronic absorption spectra were recorded on a Varian Cary 50 spectrophotometer in the range from 200 to 900 nm in the indicated solvent at room temperature.

**SQUID magnetic susceptibility and magnetization measurements.** Temperature-dependent magnetic susceptibility measurements were carried out with a Quantum-Design MPMS-XL-5 SQUID magnetometer equipped with a 5T magnet in the range from 300 to 2.0 K. The powdered samples were contained in a gel bucket and fixed in a nonmagnetic sample holder. Diamagnetic corrections of the constituent atoms for **3** were calculated from Pascal's constants[51] and found to be − 854.0 × 10$^{-6}$ cm$^3$ mol$^{-1}$. Experimental susceptibilities were also corrected for the magnetization of the sample holder (0.0001 cm$^3$ mol$^{-1}$).

**$^{57}$Fe Mössbauer spectroscopy.** $^{57}$Fe Mössbauer spectra of powdered samples and frozen aqueous solutions were recorded in transmission geometry with a $^{57}$Co source embedded in a rhodium matrix using a conventional constant-acceleration Mössbauer spectrometer ('Wissel') equipped with a nitrogen gas-flow cryostat at 80 and 293 K. Isomer shifts are given relatively to an α-Fe foil at ambient temperature. Simulations of the experimental data were performed with the Recoil program[52]. For $^{57}$Fe Mössbauer spectra of compounds in this paper, see Supplementary Figs 12,15–17.

**Electrochemical measurements.** All electrochemical measurements (cyclic voltammetry) were performed under a dry nitrogen atmosphere at 25 ± 1 °C using 10$^{-3}$ M solutions either in water or acetonitrile with 0.1 M supporting electrolyte (sodium perchlorate for aqueous, tetrabutylammonium perchlorate for acetonitrile solutions) at different sweep rates ranging from 25 to 1,000 mV s$^{-1}$ and a conventional three electrode cell with a Metrohm 6.1204.120 Platinum Unpolished Rotating Disk Electrode as a working electrode, a Metrohm 6.0343.000 platinum auxiliary electrode and a Metrohm 6.0728.020 Ag/AgCl reference electrode on a Metrohm 757 VA Computrace instrument. All the reported half-wave potentials in acetonitrile were referenced against the ferrocenium/ferrocene (Fc/Fc$^+$) redox couple (10 mM ferrocene solution).

**Single-crystal X-ray crystallographic analyses.** For X-ray analysis of compounds in this paper, see Supplementary Table 6, Supplementary Figs 6,7,18–21 and Supplementary Data 1–3.

Measurements were carried out on a Bruker SMART APEX II CCD diffractometer at 293(2) K with horizontally mounted graphite crystal as a monochromator and Mo-K$_\alpha$ radiation (λ = 0.71073 Å). Data were collected and processed using APEX 2 (ref. 53). A semi-empirical absorption correction (SADABS)[54] was applied to all data. The structures were solved by direct methods (SHELXS-97)[55] and refined by full-matrix least squares on all F$_o^2$ (SHELXL-2014/7)[56] anisotropically for all nonhydrogen atoms.

**2:** One of the methyl groups of one of the tetrabutylammoium cations was found to be disordered over two positions with occupancies of 0.71 and 0.29, respectively. The C–C bond lengths involving this methyl group (C26–C27a and C26–C27b) were restrained to ensure proper geometry using DFIX instruction of SHELXL2014 (ref. 56). To achieve reasonable anisotropic displacement ellipsoids, a further EADP instruction was applied for both positions. In one of the solvate chloroform molecule, three chlorine atoms were found to be disordered over two positions with occupancy factors 0.86 and 0.14, respectively. These occupancies were established by free refinement of both positions in anisotropic refinement. Propeller-like disorder of the chlorine atoms in the lattice chloroform molecules leads to a large difference between U$_{eq}$ parameters of well-arranged carbon atom and disordered chlorine. The two-model disordered chlorine atoms (Cl1A, Cl1B; Cl2A, Cl2B; Cl3A, Cl3B) were given the same anisotropic thermal parameters. The C–H hydrogen atoms were located from the difference Fourier map but constrained to ride on their parent atoms with C–H = 0.96–0.98 Å, and U$_{iso}$ = 1.2–1.5 U$_{eq}$ (parent atom).

**3:** One of the solvate water molecules was found to be disordered over two positions O10A and O10B with occupancies of 0.29 and 0.71, respectively. The occupancies were established by free refinement of the occupancy factors with the sum constrained to 1.0. As their anisotropic displacement ellipsoids for the disordered oxygen atoms O10a and O10B were rather elongated, DELU/SIMU restraints were also applied[56,57]. Another solvate water molecule was partially lost from the structure and therefore occupancy factor of O14W was freely refined to 0.28 in the final refinement cycle. The O–H hydrogen atoms (excluding those bonded to O14W atom) were located from the difference Fourier map, and the geometric parameters of those bonded to O4W, O5W, O6W, O7W, O8W and O10W water molecules were refined freely with U$_{iso}$ = 1.5 U$_{eq}$ (parent atom). The rest of the O—H hydrogen atoms were constrained to ride on their parent atoms with U$_{iso}$ = 1.5 U$_{eq}$ (parent atom). H atoms bonded to O14W atom were calculated with the program HYDROGEN[58]. Other hydrogen atoms were positioned geometrically and were constrained to ride on their parent atoms, with C–H = 0.93–0.97 Å, and U$_{iso}$ = 1.2 U$_{eq}$ (parent atom).

**5:** ISOR instructions of SHELXL2014 (ref. 56) were applied for C14, C15 and O7 atoms of the lattice isopropanol molecule in order to eccentric ADPs of these atoms that could not be resolved in the form of static disorder. The O–H hydrogen atoms

were located from the difference Fourier map, and their geometric parameters were refined freely with $U_{iso} = 1.5 U_{eq}$ (parent atom), with exception of the O–H atoms of isopropanol and O3W solvate water molecules that were constrained to ride on their parent atoms with $U_{iso} = 1.5 U_{eq}$ (parent atom). The C–H hydrogen atoms were positioned geometrically and were constrained to ride on their parent atoms, with C–H = 0.96–0.98 Å, and $U_{iso} = 1.2 U_{eq}$ (parent atom).

**DFT calculations.** The BP86/TZVP[59–61] method was used for optimization and Hessian calculation[62]. Single-point calculations were carried out under B3LYP/ TZVP[63,64] level of theory, the isomer shifts and quadrupole splitting parameters were then obtained from calculations on the optimized structures as well as on the crystallophycally determined geometries (Supplementary Fig. 22). Detailed description of the methods used, lists of the calculated structural, energetic, electron density and spectral parameters is given in the Supplementary Methods, Supplementary Tables 7–11 and Supplementary Data 4.

**Data availability.** Atomic coordinates and structure factors for the reported crystal structures have been deposited in the Cambridge Crystallographic Data Centre under the accession codes CCDC-1400635 (for **2**), CCDC-1400636 (for **3**) and CCDC-1458636 (for **5**). All other data that support the findings of this study are available within Supplementary Information files, and are also available from the corresponding author on reasonable request.

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

## Acknowledgements

The research leading to these results has received funding from the European Community's Seventh Framework Programme (FP7/2007-2013) under grant agreement no 295160. D.B. acknowledges the Marie Curie Individual Fellowship funding, project number 657514.

## Author contributions

S.T. and I.O.F. carried out the synthetic work and refined X-ray single-crystal structures. S.T. performed absorption spectroscopy experiments and analysed the data. S.I.S. and V.K. collected and processed magnetochemical and Mössbauer data. E.G.-K. collected ESI-MS and recorded cyclic voltammogram (CV) data. V.B. collected the X-ray data and solved X-ray single-crystal structures. D.B. performed the quantum chemical modelling. I.O.F. conceived and supervised this work and prepared the manuscript with input of all co-authors.

## Additional information

**Competing financial interests:** The authors declare no competing financial interests.

