## [Peer review file · Nature Communications]

Reviewers' comments:

Reviewer #1 (Remarks to the Author):

This is a beautiful study on a unique iron(IV) compound. The technical work is outstanding. Simple is beautiful and the simplicity of the access to this species is admirable. I support publication in Nature Communications.

However, I find the scholarship misleading. The abstract and introduction the authors have given might have been appropriate 20 years ago—the obsession with stability polarizes the article in a way that inaccurately depicts where high valent iron chemistry is today. For example, TAML systems, which use four deprotonated organic amides to give strongly reducing ferric systems, also react rapidly with molecular oxygen to give perfectly stable iron(IV) complexes that have been, as with the subject complex of this study, characterized by X-ray crystallography—many years ago. This reader would not realize this from the present article. The facts referenced that an iron(IV) TAML complex slowly “decays” in exceptionally aggressive hydrolytic conditions (pH≈14) or that Fe(IV)OFe(IV) systems are reactive (the TAML system itself does not decay here and the transformations connect with the catalytic processes TAMLs enable) in coordinating solvents is hardly something that distinguishes the subject system where the ferric species also reacts with oxygen to give iron(IV). A number of perfectly stable iron(IV)TAML complexes have been structurally characterized. So instead, wouldn't the authors agree that the fact that they have found a ferric center in the environment of six deprotonated, substituted amide ligands to be strongly reducing might be expected by projection from TAML science? Moreover, the TAML series of complexes have a rich high valent iron science and a catalytic chemistry that rivals and even out-performs the enzymes they model.

To mention environmental applications and Fenton chemistry in the introduction without reference to many publications showing TAML systems so outperform Fenton chemistry in environmental applications as to eclipse it for the future of purification of water, is to misrepresent the value to the world of high valent iron chemistry. In my opinion, this beautiful system does not need stability red herrings to distinguish it from existing systems. A more accurate analysis of the lineage of these compounds would acknowledge their connection with TAML systems.

There is one obvious question that I anticipate readers will want addressed concerning where this chemistry might go. We are shown cyclic voltammograms for only the Fe(IV/III) couple. What happens electrochemically at more oxidizing potentials? Can iron(V) be made? Can iron(VI) be made? Can even iron(VII) or iron(VIII) be made? This type of a complex could be the best system around for most easily accessing these rarer (V, VI) and even unprecedented states (VII, VIII). Or are the couples irreversible due to some stability problem? Are we going to see a series of further publications on higher valent states? This would be truly inspiring. Or does this particular system run into a stability wall? The full cyclic up to strongly oxidizing conditions should be disclosed as this is where the more interesting scientific story lies.

In conclusion, this is a beautiful system where the technical details are outstanding and it deserves publication in a prominent journal such as Nature Communications. I recommend a rewriting of the abstract and introductory section to change the misleading emphasis on stability, emphasize the more appropriate references and intellectual lineage and connected exceptional catalysis that high valent iron chemistry offers to the world today, highlight instead that this system has significantly reduced the iron(IV/III) couple compared to all other systems to give a very strongly reducing ferric system that is quite remarkable in this regard, and give some

indication of whether or not there is a further higher valent chemistry available to this system.

Reviewer #2 (Remarks to the Author):

I have been asked to assess the crystallographic components of this paper. While the structure analysis appears to be generally satisfactory, there are a considerable number of significant issues which must be addressed, mostly concerned with a lack of clarity over important procedures and decisions.

Structure 2

For structure 2 the atomic displacement parameters for the atoms N13–C19 are borderline and restraints may be advantageous. All restraints and constraints applied should be explained.

The experimental temperature quoted corresponds to the default temperature assigned by the outmoded refinement program SHELXL-97 (SHELX-2014 is freely available).

The occupancies of the atoms C11A–C13C have been fixed at values of 0.38, 0.43 or 0.19, but it is not clear how these occupancies were established. An explanation and justification are required.

There is a small discrepancy between the calculated and reported values for C27B-H27F.

The CIF contains a number of spurious valence angles such as CL3C-C13-CL2B = 36.90° which should be removed.

The high number of validation alerts relating to the structural model requires closer attention.

Structure 3

For structure 3, all the restraints applied need to be explained, at least in the supplementary data. The large number of fixed H atoms in hydrogen bonds D-H...A is unusual.

How were the occupancies of O19A/B established?

There is a minor discrepancy between the observed and calculated values for H1WA...O2.

Structure 5

In structure 5, the ratio of 'observed'/unique reflections is only 46 %. The unit cell should therefore be specifically checked for spurious lattice doubling.

There is a slightly short intermolecular D-H...H-D contact involving H2W1...H2W3: the placement of these atoms should be checked.

All restraints and constraints should be explained.

The large number of fixed H atoms in hydrogen bonds D-H...A is unusual.

The predicted and reported absorption corrections are identical: this is another example of default SHELXL-97 behaviour when no real values are supplied. The same applies to the

temperature (293 K).

A large number of meaningless torsion angles arise from linear arrangements of three of the atoms. These should be removed, as should three spurious valence angles with values of less than 45°.

Reviewer #3 (Remarks to the Author):

In this paper, the authors report the synthesis of a novel iron(IV) complex by a metal-templated assembly of oxalidihydrazide and formaldehyde in alkaline aqueous solution in the presence of air. This species with a hexaanionic clathrochelated ligand is stable under ambient conditions and crystallizes with a number of counteranions. The crystal structure shows a high-valent iron center that is encapsulated in an N₆ cage with trigonal symmetry. The iron(IV) complex is characterized by magnetic measurements to have two unpaired electrons and by cyclic voltammetry to exhibit a quasi-reversible reduction at -1.2 V vs Fc. This low potential explains its assembly under aerobic conditions, its stability, and poor oxidizing properties. It is an important addition to the handful of iron(IV) complexes that have been structurally characterized. I would recommend its publication in *Nature Communications*.

Comments

The abstract should mention that the clathrochelated complex was assembled in basic aqueous media from oxalidihydrazide and formaldehyde.

The statement about half-lives in lines 30-31 is inaccurate and contradicts the statement in lines 59-60.

The EPR signal for the low-spin iron(III) center with an unusual $dz_{z^2}^2 dx_{x^2-y^2}^1$ or $dz_{z^2}^2 dx_{x^2-y^2}^2$ configuration should be reported.

Lines 181-186: I disagree with the claim that the iron(III) complex is a strong reductant. Yes, it is readily oxidized by O₂, but this reaction alone does not qualify it for its description as a strong reductant. Also, given its redox potential and reversibility, it is not at all surprising that it would catalytically oxidize a mild reductant such as ascorbate in the presence of air.

Reviewer #4 (Remarks to the Author):

The present manuscript, titled “Indefinitely stable water- and air-born iron(IV) cage complexes”, submitted by S. Tomy *et al.*, reports unprecedented Fe(IV) hexahydrazide clathrochelate complexes that are formed in aqueous solution by air oxidation. The very remarkable stability of these high-valent iron complexes is ascribed to the 6- charge of the clathrochelate ligand and – likely more importantly – the fact that the Fe(IV) ion is encapsulated by the macropolycyclic cage. The complexes are well (and sufficiently) characterized by single-crystal x-ray diffraction, electrochemical, and (isofield) magnetization studies, (zero-field) ⁵⁷Fe Mössbauer and UV/vis spectroscopy, as well as ESI-MS spectrometry. The Fe(IV) complexes electronic structure is supported by DFT calculations

(BP86/TZVP).

Obviously, for such interesting Fe(IV) complexes one is tempted to request a more in-depth characterization of the complexes' electronic structure (such as field-dependent magnetization and Mössbauer spectroscopy), and while this reviewer would appreciate those data, I consider the presented characterization "sufficient" (see comments below). However, no high-resolution MS nor elemental analyses are provided. Whether or not this is an absolute requirement for the publication of new compounds in *Nature Communications*, I will leave to the Editor.

This is an exceedingly interesting and very timely contribution to the field of high-valent iron coordination chemistry. The report certainly is of great current relevance and is of interest to the broad readership of *Nature Communications*. Also, the manuscript is well written, clear and concise.

Accordingly, I strongly recommend this work for publication in *Nature Communications*! The following minor concerns and suggestions should be addressed prior to final acceptance.

1) The title should be revised; this reviewer can only guess what "water and airborne" is supposed to mean but this not an English (nor chemical) expression.

2) The chemistry and literature references of high-valent iron nitrido complexes have been severely neglected. With the exception of J. Smith's Fe(V) nitride, published in *Science* **2011**, the authors chose to focus on high-valent iron oxidos but have mostly neglected the fact that iron nitrides have pioneered the field of high-valent iron chemistry. In the context of Fe(IV) complexes, however, the work by J. Peters (*JACS* **2004**), K. Meyer (*Angewandte* **2008**), and J. Smith (*JACS* **2008**) must be cited, and the authors ought to acknowledge the fact that these nitridos have not been synthesized with the use of strong oxidants but *via* elimination of anthracene and dinitrogen from a coordinated amide and azide ligand, respectively. Also, these Fe(IV) complexes are quite stable in solid state and in solution! Finally, the authors ought to acknowledge the recently published Cp*₂Fe(IV)-complexes (*Science* **2016**), which are stable in solid state (under inert gas atmosphere) and in solution (HF, SO₂); albeit those solvents are quite exotic. However, the electrochemical properties of the Cp*₂Fe⁺ system have been determined by R. Gale (*J. Organomet. Chem.* **1980**) but remain undisclosed in this manuscript.

Accordingly, the sentences "Synthetic high-valent iron compounds can only be obtained with the use of oxidants, and most are highly reactive and unstable under ambient conditions. Moreover, all known high-valent iron compounds also appear to be unstable in solution (...), undergoing quick decomposition,..." is not correct and must be revised (page 2, line 28 ff).

Similarly, the second paragraph on page 3 (line 57 ff) discusses the instability of "all known high-valent iron compounds (both inorganic and coordination complexes) appear to be unstable in solution, ..." but disregards the Fe(IV) nitrides mentioned above. Last but not least, the concluding sentence "This feature clearly distinguishes the presented iron(IV) complexes from all earlier reported high-valent iron compounds which are typically unstable, highly reactive and quickly decomposing species" must be revised as well

3) page 4, line 75: typo in “hydraizide-“

4) page 4, line 80: delete “undoubtedly”

5) page 5: the complexes are synthesized “alkaline aqueous media” and authors report their remarkable stability in aqueous solution at pH 1 and pH 7 but do not comment on the complexes’ stability in basic solution, e.g., at pH 14. Please comment!

6) page 5: replace “2D-nets” with “2D-networks”

7) page 7, line 128: This reviewer wonders why the authors decided to call the observed $S = 1$ spin state for the tetravalent title complex “intermediate”. Isn’t this a standard low-spin complex (just like the reduced species is standard lowspin, $S = S$)?

8) page 8, line 142: The MÜSSBAUER isomer shift and quadrupole splitting parameters were determined and compared to “other iron(IV) compounds”; again, neglecting all Fe(IV) nitrido complexes as well as the recently reported $[\text{Cp}^*_2\text{Fe(IV)}]^{2+}$ species!

9) page 10, 2nd paragraph: ^{57}Fe MÜSSBAUER parameters of an *in situ* reduced frozen solution of tetravalent **3**, namely the Fe(III) trianion, are presented. Clearly, the observed isomer shift is more positive (as expected) but the quadrupole splitting, ΔEQ , is considerably smaller at 1.12 mm/s (compared to 2.505 mm/s for **3**), “which are characteristic for low-spin iron(III),” as stated by the authors. No references are given.

This reviewer does not agree with this statement. Generally, Fe(III) low-spin complexes have substantially larger quadrupole splitting parameters. In fact, in this particular case, and given the orbital ordering, an Fe(III) low-spin complex ought to exhibit a *larger* quadrupole splitting than its parent Fe(IV) species. Considering that the Fe(IV) species has 2 electrons in d_{z^2} and a total of 2 electrons in (d_{xy} , $d_{x^2-y^2}$), the valence contribution to the electric field gradient (EFG) is zero ! Now, adding 1 more electron to (d_{xy} , $d_{x^2-y^2}$) results in a non-zero valence contribution to the EFG for Fe(III). Assuming an approximately equal covalence contribution to the EFG, likely of opposite sign, might explain the observed quadrupole splittings for the tetra- and trivalent complexes but this ought to be discussed. The current statement, however, is not sufficient. In fact, a fairly related discussion is held in the above-mentioned $\text{Cp}^*_2\text{Fe(IV)}$ publication.

Reviewer #1 (Remarks to the Author):

(1) However, I find the scholarship misleading. The abstract and introduction the authors have given might have been appropriate 20 years ago—the obsession with stability polarizes the article in a way that inaccurately depicts where high valent iron chemistry is today. For example, TAML systems, which use four deprotonated organic amides to give strongly reducing ferric systems, also react rapidly with molecular oxygen to give perfectly stable iron(IV) complexes that have been, as with the subject complex of this study, characterized by Xray crystallography—many years ago. This reader would not realize this from the present article. The facts referenced that an iron(IV) TAML complex slowly “decays” in exceptionally aggressive hydrolytic conditions (pH≈14) or that Fe(IV)OFe(IV) systems are reactive (the TAML system itself does not decay here and the transformations connect with the catalytic processes TAMLs enable) in coordinating solvents is hardly something that distinguishes the subject system where the ferric species also reacts with oxygen to give iron(IV). A number of perfectly stable iron(IV)TAML complexes have been structurally characterized. So instead, wouldn't the authors agree that the fact that they have found a ferric center in the environment of six deprotonated, substituted amide ligands to be strongly reducing might be expected by projection from TAML science? Moreover, the TAML series of complexes have a rich high valent iron science and a catalytic chemistry that rivals and even out-performs the enzymes they model.

RESPONSE: We agree with this remark. We mentioned in the manuscript that TAML systems react with molecular oxygen to give stable iron(IV) complexes and added the corresponding reference. We appreciated that the main impetus of our study was indeed TAML chemistry and indicated this in the manuscript. The introduction and abstract have been significantly changed and rewritten accordingly, in order to remove excessive accent on stability issues.

(2) To mention environmental applications and Fenton chemistry in the introduction without reference to many publications showing TAML systems so outperform Fenton chemistry in environmental applications as to eclipse it for the future of purification of water, is to misrepresent the value to the world of high valent iron chemistry. In my opinion, this beautiful system does not need stability red herrings to distinguish it from existing systems. A more accurate analysis of the lineage of these compounds would acknowledge their connection with TAML systems.

RESPONSE: We are grateful to the referee for this useful remark. Of course, TAML complexes were the main stimulus inspiring us to develop the hexahydrazide cages. We introduced the corresponding sentence to the subsection “Reaction design”:

“In search of ligand systems that provide an extraordinarily efficient stabilization of high-valent iron, we have been inspired by TAML-based complexes, as they are among the most stable high-valent iron species reported up to date (in particular, in aqueous solution)¹²⁻²¹. The deprotonated amide groups are known to be one of the best donors for the stabilization of high oxidation states of transitional metals. Particular efficacy of TAMLs is due to both, strong σ -donor capacity and high total negative charge that can provide fully deprotonated polydentate ligands”.

We also updated the introduction, namely, its part where environmental applications and Fenton chemistry is discussed, with information on use of TAML complexes in various catalytic processes and added the corresponding references. We agree that excessive attention to stability issues is not necessary, so that we modified the introduction accordingly.

(3) There is one obvious question that I anticipate readers will want addressed concerning

where this chemistry might go. We are shown cyclic voltammograms for only the Fe(IV/III) couple. What happens electrochemically at more oxidizing potentials? Can iron(V) be made? Can iron(VI) be made? Can even iron(VII) or iron(VIII) be made? This type of a complex could be the best system around for most easily accessing these rarer (V, VI) and even unprecedented states (VII, VIII). Or are the couples irreversible due to some stability problem? Are we going to see a series of further publications on higher valent states? This would be truly inspiring. Or does this particular system run into a stability wall? The full cyclic up to strongly oxidizing conditions should be disclosed as this is where the more interesting scientific story lies.

RESPONSE: We completely agree with the reviewer that enormously unusual stability of these complexes opens a tempting perspective to obtain the clathrochelate species containing iron in higher (+5, +6..) oxidation states. We performed CV scanning towards positive potentials in acetonitrile which reveals a quasireversible wave at $E_{1/2} = 0.02$ V vs. Fc/Fc⁺ (with $\Delta E_p = 72$ mV) that probably corresponds to the Fe^{5+/4+} couple (Supplementary Figure 11). At higher potentials, a series of irreversible features (with $E_{p,a}$ at 0.57, 0.69 and 0.87 V) has been observed as well (Supplementary Figure 9). The corresponding sentence is added to the main text, two additional supplementary figures are added as well, Supplementary Table 4 is updated.

(4) In conclusion, this is a beautiful system where the technical details are outstanding and it deserves publication in a prominent journal such as Nature Communications. I recommend a rewriting of the abstract and introductory section to change the misleading emphasis on stability, emphasize the more appropriate references and intellectual lineage and connected exceptional catalysis that high valent iron chemistry offers to the world today, highlight instead that this system has significantly reduced the iron(IV/III) couple compared to all other systems to give a very strongly reducing ferric system that is quite remarkable in this regard, and give some indication of whether or not there is a further higher valent chemistry available to this system.

RESPONSE: The abstract and introductory section are rewritten according to the reviewer's recommendations.

Reviewer #2 (Remarks to the Author):

Structure 2

(1) For structure 2 the atomic displacement parameters for the atoms N13–C19 are borderline and restraints may be advantageous. All restraints and constraints applied should be explained.

RESPONSE: All the mentioned atoms belong to the lattice chloroform molecules (C13-C19) and tetrabutylammonium cation (N13). It is well known that lattice chloroform molecules are often affected by propeller-like disorder that leads to a higher U_{eq} of rotating chlorine atoms in comparison with carbon. All attempts to treat this problem using ISOR, SIMU and DELU instructions in SHELXL did not improve the situation. The detailed analysis of the corresponding alert PLAT244_ALERT_4_C assumes that false alarms may occur for the terminal groups such as the t-butyl moiety. Since both moieties, affected by this problem, are very similar to t-butyl, we assume that the atom assignment in the structure is correct since both molecules were used in the synthesis.

(2) The experimental temperature quoted corresponds to the default temperature assigned by the outmoded refinement program SHELXL-97 (SHELX-2014 is freely available).

RESPONSE: The structure was re-refined using SHELX-2014, the correct temperature is indicated.

(3) The occupancies of the atoms C11A–C13C have been fixed at values of 0.38, 0.43 or 0.19, but it is not clear how these occupancies were established. An explanation and justification are required.

RESPONSE: We are grateful to the referee for this useful remark. The occupancies of all three positions were freely refined using SUMP instruction on the early refinement stages using isotropic approximation for the disordered atoms. Simultaneous anisotropic refinement of disordered atoms and their occupancies makes it instable. Since positions A and B are very close to each other we decided to split them and refine both two positions in anisotropic approximation and their occupancy at the same time. The corresponding statement was given to the experimental part.

(4) There is a small discrepancy between the calculated and reported values for C27B-H27F.

RESPONSE: Corrected.

(5) The CIF contains a number of spurious valence angles such as CL3C-C13-CL2B = 36.90° which should be removed.

RESPONSE: Corrected.

(6) The high number of validation alerts relating to the structural model requires closer attention.

RESPONSE: The validation of the structural model using PLATON checkcif routine shows no alerts of type A and B which are normally considered as critical. There are some alerts of type C which cannot be eliminated just by refinement (see our answer on the first and the third remarks).

Structure 3

(7) For structure 3, all the restraints applied need to be explained, at least in the supplementary data. The large number of fixed H atoms in hydrogen bonds D-H...A is unusual.

RESPONSE: Since the anisotropic displacement ellipsoids for the disordered oxygen atoms O10a and O10B were rather elongated, DELU/SIMU restraints were applied. The corresponding statement was given to the experimental part.

In the resubmitted structure, the O—H hydrogen atoms geometric parameters were refined freely with $U_{\text{iso}} = 1.5 U_{\text{eq}}(\text{parent atom})$ for the O—H hydrogen atoms of O4W, O5W, O6W, O7W, O8W, O10W solvate water molecules while for the rest of water molecules the geometric parameters of the hydrogen atoms were constrained to ride on their parent atoms with $U_{\text{iso}} = 1.5 U_{\text{eq}}(\text{parent atom})$. Free refinement of the latter hydrogen atoms did not result in convergence of the refinement.

(8) How were the occupancies of O19A/B established?

RESPONSE: The occupancies of O10a and O10b were freely refined and then fixed in the final refinement cycle. In the resubmitted structure we allowed to refine the occupancies of both positions in the final refinement cycle.

(9) There is a minor discrepancy between the observed and calculated values for H1WA...O2.

RESPONSE: Corrected.

Structure 5

(10) In structure 5, the ratio of 'observed'/unique reflections is only 46 %. The unit cell should therefore be specifically checked for spurious lattice doubling.

RESPONSE: Since the structure crystallizes in P-1 space group with $z = 2$ and one molecule of the complex in asymmetric unit we do not think that lattice doubling could be a case. Otherwise the molecule will not fill into the unit cell any more. Moreover, an accurate inspection of the diffraction images does not show large amount of theoretical diffraction spots which do not fit to the observed ones. Therefore we assume that the reason of low ratio of observed/unique reflections is the poorly scattering single crystal only.

(11) There is a slightly short intermolecular D-H...H-D contact involving H2W1...H2W3: the placement of these atoms should be checked.

RESPONSE: The problem is solved after free refinement of most of the hydrogen atoms positions with SHELXL2014.

(12) All restraints and constraints should be explained.

RESPONSE: All restrains are now explained in the experimental part.

(13) The large number of fixed H atoms in hydrogen bonds D-H...A is unusual.

RESPONSE: In the resubmitted structure, the O—H hydrogen atoms geometric parameters were refined freely with $U_{\text{iso}} = 1.5 U_{\text{eq}}(\text{parent atom})$, with exception of the O—H atoms of isopropanol and O3W solvate water molecules, which were constrained to ride on their parent atoms with $U_{\text{iso}} = 1.5 U_{\text{eq}}(\text{parent atom})$. Free refinement of the latter hydrogen atoms did not result in convergence of the refinement.

(14) The predicted and reported absorption corrections are identical: this is another example of default SHELXL-97 behaviour when no real values are supplied. The same applies to the temperature (293 K).

RESPONSE: Corrected . The structure is re-refined with SHELXL-2014.

(15) A large number of meaningless torsion angles arise from linear arrangements of three of the atoms. These should be removed, as should three spurious valence angles with values of less than 45°.

RESPONSE: Corrected.

Reviewer #3 (Remarks to the Author):

(1) The abstract should mention that the clathrochelated complex was assembled in basic aqueous media from oxalidihydrazide and formaldehyde.

RESPONSE: The corresponding sentence of the abstract is reformulated as follows:

“Here we describe unprecedented iron(IV) hexahydrazide clathrochelate complexes that are assembled in alkaline aqueous media from iron(III) salts, oxalidihydrazide and formaldehyde in the course of a metal-templated reaction accompanied by air oxidation. The complexes can exist indefinitely at ambient conditions without any sign of decomposition, both in water, nonaqueous solutions and in the solid state.”

(2) The statement about half-lives in lines 30-31 is inaccurate and contradicts the statement in lines 59-60.

RESPONSE: We agree with this remark. As we significantly shortened the abstract (to fit the limit of 150 words), the corresponding statements have been omitted.

(3) The EPR signal for the low-spin iron(III) center with an unusual $dz_{z^2}^2 dx_{y^2}^2 d_{x^2-y^2}^1$ or $dz_{z^2}^2 dx_{y^2}^1 d_{x^2-y^2}^2$ configuration should be reported.

RESPONSE: We have made two independent attempts to record EPR spectra of the solutions containing iron(III) species generated by reduction of iron(IV) complexes with excess of sodium dithionite or sodium ascorbate. In both cases reduction was accomplished in aqueous solution (as we observed that it proceeds only in aqueous media, we failed to realize reduction in any other solvent), then the resulting solution was diluted with methanol or ethylene glycol to obtain aqueous-nonaqueous (1:4) mixtures for recording of EPR spectra. The spectra were recorded both at room and liquid nitrogen temperatures. In both cases no EPR signal was observed, the iron(III) species seemed to be EPR-silent.

This observation is not very surprising as for such complexes with trigonal-prismatic geometry and significant geometrical distortions a strong anisotropy in the g-tensor is expected.

[REDACTED]

-Further proof/disproof of the assumption requires additional calculations (e.g., not DFT) which are out of scope of this article. Also, there is a “technical” reason that can be considered: we used water-based solutions, for which EPR spectra were not registered. As reduction proceeds only in aqueous solution, we could not prepare non-aqueous solutions of iron(III) species (more suitable for EPR) neither by precipitation/redissolution nor by extraction.

We expect to explain why the low-spin iron(III) species are EPR-silent under the applied experimental conditions in the course of our further experimental and theoretical study of the hexahydrazide clathrochelates. EPR study of single crystals may be of importance. [REDACTED]

[REDACTED]

(4) Lines 181-186: I disagree with the claim that the iron(III) complex is a strong reductant. Yes, it is readily oxidized by O₂, but this reaction alone does not qualify it for its description as a strong reductant. Also, given its redox potential and reversibility, it is not at all surprising that it would catalytically oxidize a mild reductant such as ascorbate in the presence of air.

RESPONSE: We agree with this remark. We meant that the iron(III) clathrochelate complexes, unlike typical ferric compounds, act as reductants with respect to dioxygen, as they are oxidized quickly being exposed to the air. Accordingly, we modified the corresponding sentence:

“In contrast to all known ferric species, the iron(III) cage complexes are oxidized by oxygen in water: their brown aqueous solutions, when exposed to air, quickly recover the initial green color indicating the regeneration of iron(IV) by atmospheric oxidation”.

We agree that it is not surprising that the iron(IV) complexes can act as catalysts in reactions of atmospheric oxidation of mild reductants. We believe that it is still worth to study, and the corresponding investigations with different substrates are underway, as it is believed that in such robust, coordinatively saturated systems, outer-sphere activation mechanism can occur.

Reviewer #4 (Remarks to the Author):

Obviously, for such interesting Fe(IV) complexes one is tempted to request a more in-depth characterization of the complexes' electronic structure (such as field-dependent magnetization and Mössbauer spectroscopy), and while this reviewer would appreciate those data, I consider the presented characterization "sufficient" (see comments below). However, no high-resolution MS nor elemental analyses are provided. Whether or not this is an absolute requirement for the publication of new compounds in *Nature Communications*, I will leave to the Editor.

RESPONSE: The ESI mass-spectra have been re-recorded in HR mode. The necessary corrections have been made, and in the present version of the manuscript the HR-MS and analytical data is reported.

1) The title should be revised; this reviewer can only guess what "water and airborne" is supposed to mean but this not an English (nor chemical) expression.

RESPONSE: We agree that the title of this article is not chemical (and probably nor an English). We composed this title aiming to make it more impressive, more attractive to potential readers who are not necessarily specialists in the subject. Our working title of this paper is

Indefinitely stable iron(IV) cage complexes formed in water by air oxidation

(or: Stabilizing high-valent iron with cage ligand: indefinitely stable iron(IV) complexes formed in water by air oxidation)

And we think that it may be used for publication. On the other hand, as none of us (co-authors) is a native speaker, we will leave to the Editor to decide if the original title could be acceptable or should be changed by that proposed above.

2) The chemistry and literature references of high-valent iron nitrido complexes have been severely neglected. With the exception of J. Smith's Fe(V) nitride, published in *Science* 2011, the authors chose to focus on high-valent iron oxidos but have mostly neglected the fact that iron nitrides have pioneered the field of high-valent iron chemistry. In the context of Fe(IV) complexes, however, the work by J. Peters (*JACS* 2004), K. Meyer (*Angewandte* 2008), and J. Smith (*JACS* 2008) must be cited, and the authors ought to acknowledge the fact that these nitridos have not been synthesized with the use of strong oxidants but *via* elimination of anthracene and dinitrogen from a coordinated amide and azide ligand, respectively. Also, these Fe(IV) complexes are quite stable in solid state and in solution! Finally, the authors ought to acknowledge the recently published Cp*₂Fe(IV)-complexes (*Science* 2016), which are stable in solid state (under inert gas atmosphere) and in solution (HF, SO₂); albeit those solvents are quite exotic. However, the electrochemical properties of the Cp*₂Feⁿ⁺ system have been determined by R. Gale (*J. Organomet. Chem.* 1980) but remain undisclosed in this manuscript.

RESPONSE: We agree with this remark. The following relevant references are added:

(a) Vogel, C., Heinemann, F. W., Sutter, J., Anthon, C. & Meyer, K. An Iron Nitride Complex. *Angewandte Chemie International Edition* **47**, 2681-2684, doi:10.1002/anie.200800600 (2008).

(b) Betley, T. A. & Peters, J. C. A Tetrahedrally Coordinated L₃Fe-N_x Platform that Accommodates Terminal Nitride (Fe^{IV}: N) and Dinitrogen (Fe^I-N₂-Fe^I) Ligands. *Journal of the American Chemical Society* **126**, 6252-6254, doi:10.1021/ja048713v (2004).

(c) Scepianiak, J. J. *et al.* Structural and Spectroscopic Characterization of an Electrophilic Iron Nitrido Complex. *Journal of the American Chemical Society* **130**, 10515-10517, doi:10.1021/ja8027372 (2008).

(d) Hendrich, M. P. *et al.* On the feasibility of N₂ fixation via a single-site Fe-I/Fe-IV cycle: Spectroscopic studies of Fe-I(N₂)Fe-I, Fe-IV N, and related species. *Proc. Natl. Acad. Sci. U. S. A.* **103**, 17107-17112, doi:10.1073/pnas.0604402103 (2006).

(e) Scepianiak, J. J. *et al.* Synthesis, Structure, and Reactivity of an Iron(V) Nitride. *Science* **331**, 1049-1052, doi:10.1126/science.1198315 (2011).

(f) Malischewski, M., Adelhardt, M., Sutter, J., Meyer, K. & Seppelt, K. Isolation and structural and electronic characterization of salts of the decamethylferrocene dication. *Science* **353**, 678 (2016).

The appropriate statements on reactivity and stability of nitride complexes are added to the text of the manuscript.

2a) Accordingly, the sentences “Synthetic high-valent iron compounds can only be obtained with the use of oxidants, and most are highly reactive and unstable under ambient conditions. Moreover, all known high-valent iron compounds also appear to be unstable in solution (...), undergoing quick decomposition,...” is not correct and must be revised (page 2, line 28 ff).

RESPONSE: We agree with this remark. As we significantly shortened the abstract (to fit the limit of 150 words), the corresponding statements have been omitted.

2b) Similarly, the second paragraph on page 3 (line 57 ff) discusses the instability of “all known high-valent iron compounds (both inorganic and coordination complexes) appear to be unstable in solution, ...” but disregards the Fe(IV) nitrides mentioned above.

RESPONSE: We agree with this remark. We reformulated this sentence as follows:

“...the vast majority of known high-valent iron compounds (both inorganic salts and coordination complexes) appear to be unstable in protic solvents ...”

We think that in this new context, mentioning the Fe(IV) nitrides seem to be irrelevant, as they are “air and moisture stable” and “not reacting with water”, which implies that they cannot be obtained in aqueous solutions.

2c) Last but not least, the concluding sentence “This feature clearly distinguishes the presented iron(IV) complexes from all earlier reported high-valent iron compounds which are typically unstable, highly reactive and quickly decomposing species” must be revised as well

RESPONSE: We agree with this remark. The mentioned sentences are revised and reformulated as follows:

“The observed enormous aqueous stability clearly distinguishes the presented iron(IV) complexes from most of the earlier reported high-valent iron compounds which are typically unstable, highly reactive, and quickly decomposing species in aqueous media”.

3) page 4, line 75: typo in “hydraizide-“

RESPONSE: Corrected.

4) page 4, line 80: delete “undoubtedly”

RESPONSE: Deleted.

5) page 5: the complexes are synthesized “alkaline aqueous media” and authors report their remarkable stability in aqueous solution at pH 1 and pH 7 but do not comment on the complexes’ stability in basic solution, e.g., at pH 14. Please comment!

RESPONSE: We checked stability of aqueous solution at pH 13 and pH 12 over 30 days period and found that the summary intensity decay was less than 3%, similarly to the measurements at pH 1. The corresponding sentence of the manuscript is updated:

“UV-VIS spectral monitoring of 10^{-4} M aqueous solution of 3 at pH 7.0 demonstrated the absence of any spectral decrease over a six month period and at pH 1.0 and 13.0 the summary intensity decay was less than 3% over the course of 30 days”.

6) page 5: replace “2D-nets” with “2D-networks”

RESPONSE: Replaced.

7) page 7, line 128: This reviewer wonders why the authors decided to call the observed $S = 1$ spin state for the tetravalent title complex “intermediate”. Isn’t this a standard low-spin complex (just like the reduced species is standard lowspin, $S = \frac{1}{2}$)?

RESPONSE: We followed the tradition to define $S = 1$ iron(IV) complexes as “intermediate spin”, in which it is assumed that $S = 2$ iron(IV) species is “high spin” and $S = 0$ iron(IV) species is “low spin”. For example, in the following recent papers this terminology is used:

- 1) Meyer, S., Klawitter, I., Demeshko, S., Bill, E. & Meyer, F. A Tetracarbene–Oxoiron(IV) Complex. *Angewandte Chemie International Edition* **52**, 901-905, doi:10.1002/anie.201208044 (2013).
- 2) England, J. *et al.* An ultra-stable oxoiron(IV) complex and its blue conjugate base. *Chemical Science* **5**, 1204-1215, doi:10.1039/c3sc52755g (2014).
- 3) Sellmann, D., Emig, S., Heinemann, F. W. and Knoch, F. A Convenient Way to Novel Fe^{IV} Complexes with Mixed N/S/P Coordination Spheres and “Innocent” Ligands. *Angewandte Chemie International Edition* **36**, 1201–1203. doi:10.1002/anie.199712011 (1997).
- 4) Comba, P., Fukuzumi, S., Kotani, H. and Wunderlich, S. Electron-Transfer Properties of an Efficient Nonheme Iron Oxidation Catalyst with a Tetradentate Bispidine Ligand. *Angewandte Chemie International Edition* **49**, 2622–2625. doi:10.1002/anie.200904427 (2010).
- 5) Nam, W., Lee, Y-M. and Fukuzumi, S. Tuning Reactivity and Mechanism in Oxidation Reactions by Mononuclear Nonheme Iron(IV)-Oxo Complexes. *Accounts of Chemical Research* **47** (4), 1146-1154. doi: 10.1021/ar400258p (2014).

Also, in a series of papers dedicated to TAML-based complexes, T. Collins and co-authors described $S = 1$ iron(IV) complexes as “intermediate spin” compounds.

On the other hand, in some papers $S = 1$ iron(IV) is defined as “low spin”, e.g.:

Milsmann, C. *et al.* Stabilization of High-Valent Fe^{IV}S₆-Cores by Dithiocarbamate(1-) and 1,2-Dithiolate(2-) Ligands in Octahedral [Fe^{IV}(Et₂dtc)_{3-n}(mnt)_n]⁽ⁿ⁻¹⁾⁻ Complexes (n=0, 1, 2, 3): A Spectroscopic and Density Functional Theory Computational Study. *Chemistry – A European Journal* **16**, 3628-3645, doi:10.1002/chem.200903381 (2010).

Interestingly, that in a recent review

Hohenberger, J., Ray, K. & Meyer, K. The biology and chemistry of high-valent iron-oxo and iron-nitrido complexes. *Nat Commun* **3**, 720 (2012).

in some cases $S = 1$ iron(IV) species is defined as “low spin” and in other cases as “intermediate spin”, probably, according to the authors’ of the cited papers assignment.

We understand, that in the specific electronic configuration which occurs in the studied chlathrocholate complexes, i.e. $(z^2)^2(x^2 - y^2)^1(xy)^1$, with degenerated SOMOs), $S = 0$ cannot be realized. Formally, $S = 1$ should be defined as “low spin”. However, to our opinion, the definition of $S = 1$ iron(IV) as “intermediate spin” is more general and unambiguous, that is why we decided to use it.

8) page 8, line 142: The Mössbauer isomer shift and quadrupole splitting parameters were determined and compared to “other iron(IV) compounds”; again, neglecting all Fe(IV) nitrido complexes as well as the recently reported [Cp*₂Fe(IV)]²⁺ species!

RESPONSE: The corresponding references on Fe(IV) nitrido complexes and [Cp*₂Fe(IV)]²⁺ are added.

9) page 10, 2nd paragraph: 57Fe Mössbauer parameters of an *in situ* reduced frozen solution of tetravalent 3, namely the Fe(III) trianion, are presented. Clearly, the observed isomer shift is more positive (as expected) but the quadrupole splitting, \square EQ, is considerably smaller at 1.12 mm/s (compared to 2.505 mm/s for 3), “which are characteristic for low-spin iron(III),” as stated by the authors. No references are given. This reviewer does not agree with this statement. Generally, Fe(III) low-spin complexes have substantially larger quadrupole splitting parameters. In fact, in this particular case, and given the orbital ordering, an Fe(III) low-spin complex ought to exhibit a *larger* quadrupole splitting than its parent Fe(IV) species. Considering that the Fe(IV) species has 2 electrons in dz² and a total of 2 electrons in (dxy, dx²-y²), the valence contribution to the electric field gradient (EFG) is zero ! Now, adding 1 more electron to (dxy, dx²-y²) results in a non-zero valence contribution to the EFG for Fe(III). Assuming an approximately equal covalence contribution to the EFG, likely of opposite sign, might explain the observed quadrupole splittings for the tetra- and trivalent complexes but this ought to be discussed. The current statement, however, is not sufficient. In fact, a fairly related discussion is held in the above-mentioned Cp*₂Fe(IV) publication.

RESPONSE: We are grateful to the Reviewer for this very useful remark and helpful suggestion. Indeed, Fe(III) low-spin complexes as a rule indicate substantially larger quadrupole splitting parameters, so that we removed the statement “which are characteristic for low-spin iron(III)” with reference to quadrupole splitting. Instead, according to the Reviewer’s recommendation, we added discussion analyzing the roles of valence and covalence contribution to quadrupole splitting.

We agree that the observed ΔE_Q values for the intermediate-spin Fe(IV) complexes are surprisingly large (2.505 mm/s for **3**) compared to that observed for the reduced low-spin Fe(III) species (1.12 mm/s). Indeed, considering the valence contribution to the electric field gradient (EFG) under the particular orbital ordering one may conclude that in the case of Fe(IV) $(z^2)^2(x^2 - y^2)^1(xy)^1$ this will be zero [$2 \times (-4/7 \langle r^{-3} \rangle) + 4/7 \langle r^{-3} \rangle + 4/7 \langle r^{-3} \rangle = 0$] while in the case of Fe(III) $(z^2)^2\{(x^2 - y^2), (xy)\}^3$ this is expected to be $+ 4/7 \langle r^{-3} \rangle [2 \times (-4/7 \langle r^{-3} \rangle) + 2 \times 4/7 \langle r^{-3} \rangle + 4/7 \langle r^{-3} \rangle = + 4/7 \langle r^{-3} \rangle]$. As rightly pointed out by the Reviewer, a large parameter of quadrupole splitting for Fe(IV) can be explained solely by noticeable covalence contribution of the opposite sign (according to the DFT calculations, its sign is negative: $\Delta E_Q = - 2.505$ mm/s for **3**) due to electron density donation from the hydrazido groups into the d_{xz} and d_{yz} orbitals (the electrons in d_{z^2} , d_{xz} and d_{yz} orbitals yield a negative contribution to ΔE_Q) considering that a lattice contribution is insignificant (up to 0.3 mm/s). In the case of Fe(III) species, a positive valent contribution arises (ca. $+4 \text{ mm s}^{-1}$) caused by adding one electron into the $d_{x^2-y^2}$, d_{xy} level which would result in significantly less negative ΔE_Q value. The negative terms of covalence contribution to EFG are approximately equal for iron(IV) and iron(III) complexes (as in both corresponding 3d configurations the d_{xz} - and d_{yz} orbitals, giving rise to the negative covalence contribution, are formally empty). The estimated resulting ΔE_Q in the iron(III) complex is then a sum of the positive valence and the negative covalence contributions. Therefore, the observed ΔE_Q is more positive than in **3** but at the same time is somewhat smaller than expected due to the fact that the total ligand contributions are in reality not exactly equal (more negative in the case of iron(III) species).

It should be noted that for the Fe(IV) complexes we drew the calculated ΔE_Q directly from the components of EFG obtained in DFT calculations:

Vxx	Vyy	Vzz
0.7672	0.7752	-1.5423

$$\eta = 0.005$$

$$\Delta E_Q = -29.031 \text{ MHz} = -2.503 \text{ mm/s}$$

The calculated value (-2.503 mm/s) is in excellent agreement with the observed value (2.505(5) mm/s).

[REDACTED]

[REDACTED]	[REDACTED]	[REDACTED]
[REDACTED]	[REDACTED]	[REDACTED]

[REDACTED]

[REDACTED]

[REDACTED]

Thus, the suggested scheme envisaging the presence of the valence and covalence contributions of the opposite signs works well and corroborated satisfactory by the experimental data. The corresponding statements are introduced into the text of the manuscript.

Reviewers' comments:

Reviewer #1 (Remarks to the Author):

I am fully satisfied by the author responses to my criticisms of the manuscript and support the publication of this work as a superb addition to high valent iron chemistry.

Reviewer #2 (Remarks to the Author):

The authors have taken on board all my comments on their original paper. They have then taken a series of appropriate actions, which range from analysing and justifying unusual structural features to repeating structural refinements using modern software and superior refinement strategies. The overall effect is a paper which is technically superior with more reliable results and fewer unnecessarily distracting structural features.

Assuming the other reviewers concur, this paper can proceed to acceptance.

Reviewer #3 (Remarks to the Author):

The authors have addressed most of my concerns. I continue to be supportive of its publication in Nature Communications.

Author response to one of my original comments: The EPR signal for the low-spin iron(III) center with an unusual $d_{z^2}^2 d_{xy}^2 d_{x^2-y^2}^1$ or $d_{z^2}^2 d_{xy}^1 d_{x^2-y^2}^2$ configuration should be reported.

The authors responded that no EPR signal could be observed at 77 K, which is not surprising for many iron(III) complexes, and experiments should be carried out at 4 K. I think the EPR spectrum of this very interesting Fe(III) complex should be obtained but I would not require it for this paper. By the way, a signal at 800 G predicted by calculations would appear at $g = 7 - 8$ and would not correspond to the g_{zz} of 10 that the authors refer to. It would surprise me if the complex would be EPR silent at liquid He temperatures. Also EPR spectra of Fe(III) complexes can be obtained in frozen aqueous solution; this is routinely done by iron biochemists and observed at liquid He temperatures.

Additional comments

Page 3, line 57-59: sentence construction is awkward; please rewrite.

Page 4, line 69: which would allow the preparation of these complexes in sufficient ...

Page 4, line 72 and other places: The authors often use the term 'air oxygen', which is unusual in English; I think they mean to say 'atmospheric O₂', so I would replace 'air oxygen' with 'atmospheric O₂' in all these instances.

Page 4 line 87: and only a small subset has been obtained in aqueous ...

Page 9, line 211: Refs 26 and 47 are cited for this statement about very negative Fe(IV) potentials. The authors also cite Borovik's paper in J. Am. Chem. Soc. 2010, 132, 12188 on an Fe(IV)=O complex with a potential of -0.90 V vs Fc.

Page 9, line 218: I disagree with the authors' characterization of sulfide and ascorbate as relatively strong reductants. Only dithionite on this list fits with this description.

Page 10, line 243 ff: I do not find this observation to be particularly surprising and in fact

diminishes the impact of their paper; I would suggest that the authors delete this sentence.

I agree with Reviewer #4's comment regarding the title. The alternative titles suggested by the authors are more appropriate.

Reviewer #4 (Remarks to the Author):

Reviewer #4 provided comments to the editor only, in which they stated that they were satisfied with the revisions and recommended accept.

Point-by-point response to the reviewer's comments

Reviewer #3 (Remarks to the Author):

(1) Author response to one of my original comments: The EPR signal for the low-spin iron(III) center with an unusual $d_{z^2}^2 d_{xy}^2 d_{x^2-y^2}^1$ or $d_{z^2}^2 d_{xy}^1 d_{x^2-y^2}^2$ configuration should be reported.

The authors responded that no EPR signal could be observed at 77 K, which is not surprising for many iron(III) complexes, and experiments should be carried out at 4 K. I think the EPR spectrum of this very interesting Fe(III) complex should be obtained but I would not require it for this paper. By the way, a signal at 800 G predicted by calculations would appear at $g = 7 - 8$ and would not correspond to the g_{zz} of 10 that the authors refer to. It would surprise me if the complex would be EPR silent at liquid He temperatures. Also EPR spectra of Fe(III) complexes can be obtained in frozen aqueous solution; this is routinely done by iron biochemists and observed at liquid He temperatures.

RESPONSE: We agree with this remark. Indeed, EPR measurements carried out at 4 K are expected to give spectra for such distorted systems, in particular, in aqueous solution. [REDACTED]

[REDACTED] We plan to perform it in our future work with participation of research groups having experience in this technique and an access to the appropriate equipment. We agree that a signal at 800 G would correspond not to the g_{zz} of 10 but of 8.50 ($g_{\text{eff}} = h\nu/\beta H = 6801.9/800 = 8.50$ at $\nu = 9,520$ GHz). If $g_{zz} = 10$ one can expect a signal at $6801.9/10 \approx 680$ G).

Accordingly, we added the following two sentences to the manuscript:

“Interestingly, the reduced solutions containing iron(III) species appeared to be EPR-silent both at room and liquid nitrogen temperatures which is not surprising for many iron(III) complexes. Evidently, registration the EPR signal for these low-spin iron(III) complexes with an unusual configuration $d_{z^2}^2 d_{xy}^2 d_{x^2-y^2}^1$ or $d_{z^2}^2 d_{xy}^1 d_{x^2-y^2}^2$ awaits future low temperature experiments at liquid helium temperatures.”

(2) Page 3, line 57-59: sentence construction is awkward; please rewrite.

RESPONSE: We agree with this remark. The corresponding sentence of the manuscript is reformulated:

“Although many reports dedicated to bioinspired oxo ($Fe^{IV,V}=O$), nitrido ($Fe^{IV,V}\equiv N$) and imido ($Fe^{IV}=N-R$) iron compounds have been published since 2000^{8,12,13,23-34} new examples of non-biomimetic high-valent iron complexes are rare and remain much less explored³⁵⁻³⁷.”

(3) Page 4, line 69: which would allow the preparation of these complexes in sufficient

RESPONSE: Corrected.

(4) Page 4, line 72 and other places: The authors often use the term ‘air oxygen’, which is unusual in English; I think they mean to say ‘atmospheric O₂’, so I would replace ‘air oxygen’ with ‘atmospheric O₂’ in all these instances.

RESPONSE: Corrected.

(5) Page 4 line 87: and only a small subset has been obtained in aqueous ...

RESPONSE: Corrected.

(6) Page 9, line 211: Refs 26 and 47 are cited for this statement about very negative Fe(IV) potentials. The authors also cite Borovik’s paper in J. Am. Chem. Soc. 2010, 132, 12188 on an Fe(IV)=O complex with a potential of -0.90 V vs Fc.

RESPONSE: We agree with this remark. This citation is added.

(7) Page 9, line 218: I disagree with the authors’ characterization of sulfide and ascorbate as relatively strong reductants. Only dithionite on this list fits with this description.

RESPONSE: We agree with this remark. We reformulate this description as “*relatively strong and moderate reductants*”.

(8) Page 10, line 243 ff: I do not find this observation to be particularly surprising and in fact diminishes the impact of their paper; I would suggest that the authors delete this sentence.

RESPONSE: We agree with this remark. The corresponding sentence is deleted.

(9) I agree with Reviewer #4’s comment regarding the title. The alternative titles suggested by the authors are more appropriate.

RESPONSE: We agree with this remark. The title is changed:
Indefinitely stable iron(IV) cage complexes formed in water by air oxidation